# Isoxazole/Isoxazoline Skeleton in the Structural Modification of Natural Products: A Review

**DOI:** 10.3390/ph16020228

**Published:** 2023-02-02

**Authors:** Xiyue Wang, Qingyun Hu, Hui Tang, Xinhui Pan

**Affiliations:** Laboratory of Xinjiang Phytomedicine Resource and Utilization, Ministry of Education, School of Pharmaceutical Sciences, Shihezi University, Shihezi 832002, China

**Keywords:** isoxazole, isoxazoline, natural products, pharmacological activity, structure-activity relationship

## Abstract

Isoxazoles and isoxazolines are five-membered heterocyclic molecules containing nitrogen and oxygen. Isoxazole and isoxazoline are the most popular heterocyclic compounds for developing novel drug candidates. Over 80 molecules with a broad range of bioactivities, including antitumor, antibacterial, anti-inflammatory, antidiabetic, cardiovascular, and other activities, were reviewed. A review of recent studies on the use of isoxazoles and isoxazolines moiety derivative activities for natural products is presented here, focusing on the parameters that affect the bioactivity of these compounds.

## 1. Introduction

Isoxazole is a heterocyclic compound with a five-membered ring that has oxygen and nitrogen atoms at the 1 and 2 positions, and their partially saturated analogs are known as isoxazoline. Many biologically active products contain derivatives of these heterocyclic compounds [1,2,3]. Derivatives containing isoxazole/isoxazoline fragments possess biological activities such as anticancer [4,5], anti-inflammatory [6,7], antibacterial [8,9], anti-Alzheimer’s disease [10,11], antioxidant [12,13], insecticidal [14], antifungal [15,16], and antidiabetic [17,18]. Isoxazolines and isoxazoles have unique electron-rich aromatic structures and have received much attention [19,20]. They make potential candidates for ring cleavage because of their weak nitrogen–oxygen bonds and aromatic character. Thus, isoxazoles and isoxazolines are particularly valuable intermediates in numerous synthetic methods of bioactive chemicals because this isoxazole ring system makes it easy to modify the substituents in their ring structures [5]. Their unusual architectures enable high-affinity binding to many targets or multiple distinct receptors, which aids in the development of innovative medications with original therapeutic applications. Therefore, chemists have been interested in developing and testing isoxazole- and isoxazoline-containing compounds with a variety of medicinal benefits [21].

In the drug discovery process, natural products play a significant role. Biologically active natural products can be obtained from plants, marine organisms, or microorganisms, and they are a vital source of drug discovery [22]. However, most directly extracted and isolated unmodified natural products cannot be directly used for clinical treatment of diseases due to the low pharmacological activity or excessive adverse effects of most natural products. Structural modifications can be used to improve the physicochemical properties and biological activity to reduce the adverse effects and improve the drug selectivity of natural products. They may also exhibit completely different biological activities from the parent [23]. Given the importance of the isoxazole/isoxazoline backbone in natural products and the remarkable bioactivity, an increasing number of studies have reported the modification of natural products by isoxazole/isoxazoline rings [24]. A number of research groups have intensively studied this field; therefore, this paper presents the progress of this field in conjunction with the latest literature reports. Modifying natural products will be addressed in this review. It will provide new conceptual approaches and directions for future research.

## 2. Biological Effects of Natural Products Containing the Isoxazole/Isoxazoline Moiety

### 2.1. Antitumour Activity

Malignant tumors are extremely complex, and they can have a major impact on people’s life and health. The incidence of cancer has been on the rise, and despite the availability of many drugs and treatments for cancer, it remains one of the greatest threats to human health. Natural products or their derivatives make up over 65% of all anti-cancer medications. Natural products thus serve a crucial clinical role in the treatment of cancer. As biosynthetic technology has advanced, A growing number of natural products are being developed for cancer therapy as clinical candidates [25]. As a result, scientists have created several isoxazole and isoxazoline derivatives with anti-cancer properties based on natural products.

Maslinic acid (MA)(Figure 1) and oleanolic acid (OA) (Figure 2) can be isolated from the natural *Olea europaea* L. MA as well as OA have been found to have anti-cancer and anti-inflammatory properties. A number of isoxazole-containing pentacyclic triterpene derivatives were created and examined by Chouaïb and coworkers. The majority of the isoxazoles, especially those generated from MA, showed remarkable anti-cancer activity in tests on the cancer cell lines EMT-6 and SW480. Isoxazole derivatives of MA **1a**, **1b**, **1c**, and **1d** (Figure 1) showed better anti-cancer properties against SW480 cell line compared to the starting substrate MA (viability (%/control) = 9, 9, 10, and 10%, respectively, 30 µM). However, only compounds **1d** and **1c** showed higher activity than MA against EMT6 (breast) (viability (%/control) = 5 and 64%, respectively, 10 µM). The scientists also performed an anti-proliferative assessment on the cancer cell lines EMT-6 and SW480 and documented the synthesis of OA isoxazole derivatives. However, according to in vitro cytotoxicity testing, OA contains a more potent anti-proliferative active compound than its isoxazole derivatives [26]. In another study, A number of OA nitrogen heterocyclic derivatives with nitrogen heterocycles at C-2 and C-3 were created by Mallavadhani et al. According to the study, pyrimidine derivatives had much greater activity than isoxazoles derivative **2** (Figure 2) against seven cancerous cell lines. The pyrimidine derivatives stopped the cell cycle and caused apoptosis in MCF cells during the S phase, according to the flow cytometric study [27].

*Streptomyces* sp. extract vegfrecine (Figure 3) as a VEGF receptor tyrosine kinase inhibitor. It exhibits potent in vitro inhibitory activity against VEGFR-1 and VEGFR-2 tyrosine kinases by blocking VEGFR-1 signaling which inhibits pathological angiogenesis associated with cancer and tumor metastasis. Adachi et al. synthesized a natural quinone compound containing an isoxazole ring through the structure of vegfrecine. Compound **3** (Figure 3), with a quinone ring thickened with an isoxazole ring, exhibited moderate inhibitory activity against VEGFR-1 tyrosine kinase (IC_50_ = 0.65 µM) and reduced inhibitory activity against VEGFR-2 tyrosine kinase compared to vegfrecine (IC_50_ = 7.1 µM). It is explained by the structure–activity relationship (SAR) that the isoxazole ring formed by the aminocarbonyl and amino groups which inhibits VEGFR-1 and VEGFR-2 tyrosine kinases by fixing the orientation of the aminocarbonyl and amino groups [28].

Tyrosol (Figure 4) is a natural phenolic compound obtained from various plants. In the study by Aissa et al., 3,5-disubstituted isoxazole derivatives (**4a–e**) (Figure 4) were synthesized from tyrosol, of which compounds **4c**, **4b**, and **4a** showed the greatest antiproliferative properties with IC_50_ values of 67.6 µM, 42.8 µM, and 61.4 µM, respectively. The derivative **4c** was superior to the positive drug than temozolomide (IC_50_ = 53.85 µM). The newly synthesized compounds exerted anticancer activity by inducing apoptosis in U87 cells. It was found that methyl, methoxy, or chloride substitutions on the R group of isoxazole derivatives enhanced their activity against U87 cells based on SAR studies [29].

A flavonoid derived from the seeds of *Hydnocarpus wightiana* Blume is called hydnocarpin (Hy) (Figure 5). It was reported that Hy exhibit antitumor effects [30]. Arya et al. modified Hy as the structural basis by introducing isoxazole rings to develop a series of new compounds. The new derivatives induce apoptosis and arrest the cell cycle at G2/M and S phases in human metastatic melanoma (A375) and human lung adenocarcinoma (A549) cells. One of the most potent compounds was compound **5** (Figure 5), which inhibited A375 at IC_50_ values of 3.6 and 0.76 µM at 24 h and 48 h, respectively, about 18–60 fold higher than Hy. In conclusion, the results indicate that attaching isoxazole to naturally occurring hydnocarpin enhanced the selectivity and cytotoxicity of the derivatives against A549 and A375 cells [31].

Forskolin (Figure 6) is a natural product of the labdane diterpene. Its antiproliferative activity was shown to be mediated by the tumor suppressor protein p53. Therefore Burra et al. introduced new isoxazoles by 1,3-dipole cycloaddition reactions at C1-OH of forskolin and tested their activity against breast cancer cell lines. The parental forskolin was active against MCF-7 cells with an IC_50_ of 63.3 µM, but did not exhibit any anticancer activity against BT-474 cells (IC_50_ > 100 µM). Among the derivatives tested in this study, the compound **6** (Figure 6) with an acetyl group at the 7th position displayed the highest activity against MCF-7 and BT-474 cell lines, exhibiting an IC_50_ of 0.5 µM [32].

A combretastatin A-4 (CA4) (Figure 7) isolated from the bark of the African willow tree *Combretum caffrum* is being developed as a potent natural cytostatic agent that blocks and apoptoses cancer cells in the G2/M phase. According to SAR studies, the *cis* double bonds bound to 3,4,5-trimethoxyphenyl ring A as well as the 4-methoxyphenyl ring B are essential to CA4’s anti-mitotic microtubule destabilizing activity. Using in vivo sea urchin embryo experiments to examine the isoxazole ring modification to CA4, Chernysheva et al. came to the conclusion that CA4 showed anti-mitotic micro-tubule destabilization at a minimum effective concentration (MEC) of 0.002 µM. Chernysheva et al. modified CA4 by introducing isoxazole rings and evaluated them using in vivo sea urchin embryo assays and concluded that CA4 exhibited anti mitotic microtubule destabilization at a minimum effective concentration (MEC) of 0.002 µM, while isoxazole derivatives **7** and **9** (Figure 7) caused altered division in sea urchin embryos at 0.005 µM and 0.02 µM, respectively [33]. In the same year, a new series of CA4 derivatives, including diaryl pyrazoles, isoxazoles, and pyrroles, were reported in the literature. To evaluate the antimitotic efficacy of their drugs against microtubules, a panel of human cancer cells and in vivo sea urchin embryo experiments were performed. The strongest antimitotic agent among the isoxazole derivatives was discovered to be compound **8** (Figure 7) (EC = 0.001 µM), possessing better antiproliferative activity than CA4 (EC = 0.002 µM) while also exhibiting comparable cytotoxicity against human cancer cells. According to structure–activity relationship studies, in the 4,5-diarylisoxazole series, removing the 3-hydroxyl group from ring B decreases the antimitotic activity, and the 3-hydroxyl group is required for antiproliferative activity [34]. Silyanova and colleagues found that only the isoxazole heterocycle and the unsubstituted benzene ring next to the heteroatom conferred the appropriate conformation of the molecule to exert antiproliferative effects through the microtubule destabilization mode of action. The 4,5-diarylisoxazoles showed greater antimitotic activity than 3,4-diarylisoxazoles [35]. In 2018, isoxazole chalcone derivatives with structural similarities to CA4 were synthesized. According to the results, the new synthesized compounds **10a** and **10b** (Figure 7) exhibited potent cytotoxic activity against DU145 prostate cancer cell lines with IC_50_ values of 0.96 µM and 1.06 µM, respectively, compared to the positive control (IC_50_ =4.10 µM). According to structure–activity relationship studies, electron-giving on the benzene ring groups, such as methoxy substituents, enhanced the anticancer activity [36].

Indirubin is an active ingredient in Chinese medicine formulas with good anti-cancer properties. Meisoindigo (Figure 8) is derived from Indirubin and is effective against cancer. Therefore, Tang et al. synthesized a series of 3-subunit indoleacetamides using meisoindigo as a structural template. Different cancer cell lines were tested by researchers to see if they were cytotoxic. Such compounds arrest the cell cycle in the G1 phase and subsequently trigger cystatinase-dependent apoptosis. Compound **11** (Figure 8) showed the best activity in the series with IC_50_ values of 2.3, 2.7, 2.2, 3.6, and 3.6 µM against MCF-7, Hep3B, KB, SF-268, and MKN-48 cancer cell lines, respectively. In comparison with meisoindigo, compound **11** had better antiproliferative properties against these five cancer cell lines (IC_50_ = 25.5, 9.0, 19.2, 37.0, 36.7 µM). As a result of adding isoxazole to precursor compounds, their antiproliferative properties were enhanced [37].

Dai et al. identified many novel bromotyrosine-derived compounds from the *Indonesian sponge*. The isolated compound purpuramine N (Figure 9) was modified by oxidation of aromatic groups to produce derivative **12** (Figure 9) containing an isoxazoline fraction. Unfortunately, NIH3T3 cells (normal mouse fibroblasts) were inhibited by compound **12**, so this compound was not further investigated. However, further studies could be conducted with compound **12** due to its moderate inhibition of aspartate protease BACE1 (memapsin-2) [38].

There are several biological activities associated with *bis*-indole alkaloids, which are sponge metabolites. Several novel *bis*-indolyl-isoxazoles and *bis*-indolyl-furans were synthesized and evaluated as antitumor agents in 10 human tumor cell lines, according to Diana et al. The most potent compound among the isoxazole derivatives was **13** (Figure 10), displaying mean IC_50_ values of 53.2 µM. Compound **13** was detected to have selective activity against A549 and LXFA 629L (lung) and UXF 1138L (uterine body). The results indicated that the bisindolyl isofuran derivatives exhibited more significant antitumor activity than isoxazole derivatives against human tumor cell lines [39].

(R)-Carvone belongs to a group of monoterpenes that are present in many natural products and bioactive molecules. An array of derivatives of monoterpenes was synthesized by Fawzi et al. Through MTT assays, all synthesized molecules were evaluated for cytotoxicity against HT-1080, A-549, MCF-7, and MDA-MB-231 cells, which concluded that compound **15** (Figure 11), an isoxazole-pyrazole heterodimer, had no significant activity against all selected cancer cell lines without significant activity. In terms of growth inhibition, compound **14** (Figure 11) was the strongest with IC_50_ values of 22.47, 25.87, 19.19, and 20.79 µM, respectively. Analyses of the SAR process revealed that the two isoxazoline parts of compound **14** are responsible for the cytotoxicity of the compound on human cancer cells. Further studies by flow cytometry showed that compound **14** caused MCF-7 cancer cells and MDA-MB-231 cancer cells to arrest in the S and G2/M phases of the cell cycle, as well as induced early apoptosis of MCF-7 and MDA-MB-231 through caspase-3/7 activation [40]. In research by Oubella and colleagues, chiral isoxazolines and pyrazole derivatives with monoterpene backbones were efficiently synthesized from (R)-Carvone. Human HT1080, MCF-7, and A-549 cancer cells were used as test subjects for the newly synthesized monoterpene isoxazoline and pyrazole derivatives’ cytotoxic properties. Among them, isoxazoline derivatives **16a**, **16b**, and **16c** (Figure 11) showed the best anticancer activity against HT1080 cells with IC_50_ values of 16.1 µM, 10.72 µM, and 9.02 µM, respectively. In contrast, pyrazole derivatives were less active in HT1080 cells, all with IC_50_ values over 100 µM. In HT-1080 cells, isoxazoline derivatives exhibited a greater anticancer activity than pyrazole derivatives [41].

(–)-*α*-Santonin is a sesquiterpene lactone compound derived from various Asian plants. Recently, it was shown by flow cytometry that naturally occurring santonin can cause G2/M phase arrest of SK-BR-3 cancer cells in the cell cycle while inhibiting the expression of cell cycle proteins A and B1 and also exerts anticancer effects by blocking the Raf/MEK/ERK pathway in breast cancer cells [42]. In a recent study, Khazir et al. synthesized new spirocyclic derivatives of the human santonin and tested their anticancer activity against cancer cell lines Among them, spiroisoxazoline derivative **17** (Figure 12) showed good activity against MCF-7 and A549 cell lines with IC_50_ values of 0.02 and 0.2 µM, respectively. As a result, compound **17** is promising as a new anticancer drug [43].

Harmine can be extracted from natural *Peganum harmala* seeds. According to research, it has significant cytotoxic activity against cancer cell lines and can induce the G2/M cell cycle arrest in breast cancer cells by regulating MAPK and AKT/FOXO3a signaling pathways [44]. Harmine was used as a scaffold to generate derivatives containing isoxazoline, and its cytotoxicity against MCF7 breast cancer and HCT116 colon cancer was assessed. As a result of the tests, Harbin had potent cytotoxicity on both cells with IC_50_ values of 0.7 µM and 1.3 µM, respectively. Among the synthesized derivatives, derivative **18** (Figure 13) with a benzene ring in the isoxazoline part showed the best activity with IC_50_ values of 9.7 µM and 0.2 µM, respectively. According to the report, the cytotoxic activity of the derivative isoxazoline part decreases when the aromatic system of the derivative bears’ methyl, methoxy, and Cl atoms in the para position, respectively [45]. Next, the group synthesized isoxazole derivatives from harmine and evaluated ovarian cancer (OCVAR-3), breast cancer (MCF-7), and colon cancer (HCT 116) cell lines using MTT assays. Among the synthesized derivatives, compound **19** (Figure 13) showed the best activity with IC_50_ values of 5.0, 16.0, and 5.0 μM, respectively. It can be seen that the isoxazoline derivatives showed more bioactivity in the colon cancer (HCT 116) cell line [46].

Plants contain betulin, which is a pentacyclic triterpene naturally occurring in many species. Lugiņina et al. synthesized dense heterocyclic derivatives based on this natural compound. The isoxazole ring was joined to the betulin scaffold by altering the triterpene ring A. Using the MTT assay, the cytotoxic activity of each derivative was evaluated against the human cancer cell lines RD TE32, A549, MS, HEp-2, and HCT 116. Among them, the N-acetyl triazole of betulin showed the strongest activity with IC_50_ 2.3–7.5 µM, whereas the thickened isoxazole ring derivative **20** (Figure 14) was not as active as the triazole derivatives with IC_50_ values of 7.9–22.1 µM. The synthetic and cytotoxic activity of betulin derivatives containing the isoxazole fraction was reported by Lugiņina et al. Five tumor cell lines were used to test the effectiveness of all derivatives (A-549, MDA-MB-231, MCF-7, KB, and KB-VIN). Compound **21a** (Figure 14) showed GI_50_ values of 11.05 ± 0.88 µM against the lung cancer cell line A549. Compounds **21b** and **21c** (Figure 14) were found against breast cancer MCF7 cells (11.47 ± 0.84 µM) and (14.51 ± 1.42 µM). Compound **21d** (Figure 14) was found on cells of breast cancer MCF7 (12.49 ± 1.18 µM) and lung cancer A549 (13.15 ± 1.56 µM). The constitutive relationship indicated that compounds with hydrophilic substituents on the isoxazole ring had stronger cytotoxic activity [47].

AD-1 (Figure 15) is a novel ginsenoside discovered to induce G0/G1 cell cycle arrest, apoptosis, and ROS production. Ma et al. introduced various heterocycles containing nitro groups at the C-2 and C-3 positions to synthesize AD-1 derivatives. The AD-1 IC_50_ value of 14.38 μM for isoxazole derivative **22** (Figure 15) showed solid cellular activity, which enriched our approach to studying the synthesis of AD-1 derivatives [48]. Smirnova et al. synthesized a *Dipterocarpus alatus* derivative **23** containing the isoxazole fraction and evaluated its cholinesterase inhibitory activity and cytotoxicity. Compound **23** (Figure 16) was found to be cytotoxic to MCF7 (breast cancer) (EC_50_ = 16.2 µM) and mildly inhibited the enzyme AChE with 15.9% inhibition. This study provides some reference for the structural modification of *D. alatus* [49].

The roots of a few wild yams contain large amounts of diosgenin. Diosgenin has received a lot of attention as a possible anticancer drug in recent years. Yildiz et al. synthetically modified the diosgenin backbone to obtain some new derivatives. The findings demonstrated that among these compounds, The pyridine-containing compound had the highest efficacy against human breast cancer (MCF-7), with an IC_50_ value of 5.72 µM. Compound **24** (Figure 17), containing the isoxazole fraction, also showed potent anticancer activity against human breast cancer (MCF-7) and lung adenocarcinoma (A549) with IC_50_ values of 9.15 ± 1.30 µM and 14.92 ± 1.70 µM, which was superior to the parent compound diosgenin (IC_50_ = 26.91 ± 1.84 µM and 36.21 ± 2.42 µM). It can be inferred that the A-ring substituted isoxazole and pyrazole fractions may enhance the anticancer activity of diosgenin derivatives [50].

In an in vitro MTS experiment, Lingaraju et al. synthesized coumarin-isoxazoline adducts and assessed their capacity to cause cytotoxicity in human melanoma cancer cell line (UACC 903) and fibroblast normal cell line (FF2441). The findings revealed that compounds **25** and **26** (Figure 18) exhibited better cytotoxicity against UACC 903, with IC_50_ values of 1.5 µM and 4.5 µM, respectively. Compound **26** was the most active, which may be due to the presence of chlorine and fluorine substitutions in the *ortho*-positions of the phenyl ring of isoxazoline. Because compound **25** has 3,4-dimethoxy on the benzene ring of the isoxazoline ring, it is more selective for melanoma cancer cells than normal cells, which may explain why compound **25** was the leading contender in this series [51]. Another team synthesized isoxazoline/isoxazole fused coumarin analogs and evaluated their cytotoxicity against human colorectal cancer (Colo-205), human hepatocellular liver cancer (HepG2) and human cervical cancer (HeLa). The findings revealed that compound **27** (Figure 18) had more sensitive activity against the HepG2 cell line than Colo-205 and HeLa cell lines, and it had the highest anti-proliferative activity (IC_50_ ≤ 50 µM) against all cell lines [52]. (–)-Deltoin is a coumarin-containing natural product extracted from flowers of *Ferula lutea* (Poir.). As potential anticancer agents based on (–)-deltoin, Znati et al. synthesized coumarin derivatives containing isoxazoline backbones. When it came to the human colon cell line HCT-116, compound **28** (Figure 18) was the most effective (IC_50_ = 3.3 µM), four times as effective as the parent chemical (IC_50_ = 14.3 µM). This finding suggests that the introduction of the isoxazoline fraction yielded better anticancer potential [53].

A significant family of compounds known as C-glycosides is present in many natural product architectures and exhibits a wide range of biological activities. Compound **29** (Figure 19), with an IC_50_ value of 0.67 M, showed the greatest cytotoxicity against MCF-7 breast cancer cells among a series of C-glycoside-linked pyrazoline and isoxazole derivatives synthesized by Kumari et al. It can be seen that pyrazoline partially favors the C-glycoside derivatives in COX-2 enzyme inhibition. The addition of the isoxazole ring to the pyrazoline structure further improves the compounds’ biological activity [54].

The spiro-pyrrolidine-oxindole ring system has specific structural properties and potent biological activities. A number of isoxazole derivatives of spiroazolidine-oxindole were created by Liu et al. Using an MTT assay, and they assessed the derivatives’ cytotoxic effects on human leukemia cell K562, human prostate cancer cell PC-3, and human lung cancer cell A549. The outcomes demonstrated that compound **30** (Figure 20) exhibited considerable cytotoxicity against these three cell lines, K562, A549, and PC-3, with IC_50_ values of 10.7 µM, 21.5 µM, and 13.1 µM, respectively. Isoxazole was added to the sporozoite–oxindole complex, and it inhibited cancer cell proliferation as well as or better than cisplatin (up to 2.1-fold) [55].

It was reported that a series of isoxazole derivatives were synthesized using naturally occurring andrographolide as a backbone by Mokenapelli et al. The cytotoxicity of the derivatives was also evaluated against HCT15, HeLa, and DU145 cell lines. Compounds **31a**, **31b**, **31c**, **31d**, and **32** (Figure 21) exhibited significant cytotoxicity against the three cancer cell lines with IC_50_ values below 40 µg/mL. Therefore, isoxazoline derivatives of andrographolide at the C-14 position are promising as anticancer agents [56].

Eugenol is a naturally occurring phenolic monoterpene obtained from clove oil having a variety of biological properties. Using eugenol as a scaffold, Oubella et al. created 1,2,3-triazole mixed isoxazoline derivatives. They next tested and assessed the derivatives’ in vitro anticancer activity against the fibrosarcoma (HT-1080), breast cancer (MCF-7 and MDA-MB-231), and lung cancer (A-549) cell lines. The mixed compounds **33a–d** (Figure 22) exhibited more significant cytotoxicity than the triazole derivatives (IC_50_ = 15–29 µg/mL) against the three cancer cell lines. According to preliminary structural investigations, the simultaneous presence of 1,2,3-triazole and isoxazole produced stronger anticancer effects. Follow-up studies showed that the most potent compound **33a**, induced apoptosis through the activation of caspase-3/7, leading to cell cycle arrest in A-549 cancer cells in the G2/M phase [57].

Sclareo is a natural product with anticancer activity. As one of the isoxazoline derivatives of sclareol, derivative **34a** (Figure 23) showed the strongest anticancer activity with cytotoxic activity (IC_50_ = 13.20–21.16 µM) against human hepatocellular carcinoma (HepG2), human cholangiocarcinoma (HuCCA-1) and human lung adenocarcinoma (A549) cell lines. When compared to the natural parent chemical sclareol, the findings demonstrated that derivative **34a** increased cytotoxicity against cancer cell lines (IC_50_ = 49.89–70.40 µM). This could be due to the structure of the derivative showing a significant hydrophobicity (due to the aliphatic skeleton) and hydrogen bonding ability (–OH), leading to increased cellular uptake of the compound, resulting in cytotoxicity [58].

Pratap et al. synthesized a series of artemisinin derivatives containing spiroisoxazoline. They evaluated the antiproliferative activity of the newly synthesized derivatives against the human lung cancer cell line (A-549), human colon cancer cell line (HCT-15), and human liver cancer cell line (Hep-G2) by MTT assay. Among all compounds, compound **35a** (Figure 24) was found to show significant cytotoxicity against all selected cell lines with IC_50_ values of 32.43, 4.04, and 46.30 µM, respectively. it can be seen that compound **35a** was more sensitive against colon cancer cell line (HCT-15). Compound **35a** containing the spiroisoxazoline fraction (IC_50_ = 4.04 µM) was nine times more active against HCT-15 cell line than the positive drug 5-fluorouracil (IC_50_ = 35.53 µM). Follow-up DNA cell cycle analysis showed that **35a** inhibited cell proliferation in the G2/M phase. It was also found that compound **35b** (Figure 24) had significant activity against *P. falciparum* (IC_50_ = 0.1 µM) [59].

Bromopyrrole alkaloids are an essential family of marine alkaloids with a wide range of biological activities. Rane et al. synthesized a series of isoxazole-containing bromopyrrolidine alkaloid derivatives and evaluated their in vitro antiproliferative activity against five human cancer cell lines by MTT assay. Among them, compound **36a** (Figure 25) exhibited the most potent anti-cancer activity. It was able to selectively inhibit oral cancer cell line KB403 with an IC_50_ of 2.45 µM, whereas compound **36b** (Figure 25) (IC_50_ = 16.58 µM) was found to be selectively cytotoxic against colon cancer cells CaCO2. Therefore, introducing brominated pyrroles into isoxazoles could enhance the anticancer activity of such compounds [60].

Methyl *β*-orsellinate is a highly functionalized natural phenolic molecule with a variety of biological properties that is present in plants. A series of isoxazole derivatives of methyl *β*-orsellinate were synthesized by Reddy et al., and four human cancer cell lines as well as the normal cell line HEK-293T (embryonic kidney) were evaluated for their antiproliferative activity in vitro. The majority of these artificial compounds demonstrated antiproliferative efficacy. The most effective combination was found to be compound **37** (Figure 26), whose IC_50_ against the MCF-7 breast cancer cell line was found to be 5 times higher (IC_50_ = 7.9 ± 0.07 µM) than the parent compound (IC_50_ = 46.63 ± 0.11 µM). The constitutive relationship analysis indicated that chlorine substitution at the benzene ring para position gave the compound a better anticancer potential. Compound **37** was shown by flow cytometric analysis to induce apoptosis and arrest the cell cycle in the G2/M phase [61].

The mono-acetate of goniodiol-7-monoacetate was obtained from ethyl acetate extract of *Goniothalamus wynaadensis* Bedd. Goniodiol diacetate was transformed into a new isoxazoline derivative by Talimarada et al. The MTT test was used to measure the derivatives’ cytotoxic activity against the human cancer cell lines MDA-MB-231, SKOV3, PC-3, and HCT-15, as well as the normal human cell line HEK 293. All isoxazoline derivatives were inhibiting cancer cells (EC_50_ < 10 µM) without damaging normal cell lines, with compound **38** (Figure 27) showing the strongest activity, compared to the positive control drug vincristine (EC_50_ = 9.02 µM, 7.00 µM), and compound **38** (EC_50_ = 6.83 µM, 6.88 µM) on SKOV3 and MDA-MB-231 cell lines exhibited better cytotoxicity. The data suggest that the presence of saturated lactones is essential for activity and that changes in the electron-absorbing or electron-donating groups on the aryl rings of all derivatives have little effect on enhancing cytotoxicity. Further molecular biology studies showed compound **38** stalled the cell cycle in the S phase [62].

A number of spiroisoxazoline derivatives based on the natural substance 1-hydroxy alantolactone have recently been created by Tang et al. Among them, compound **39** (Figure 28) exhibited the most potent antitumor activity with IC_50_ values of 2.7-5.1 µM against HeLa, PC-3, HEp-2, and HepG2 cells, respectively, which were superior to the parent compound 1β-hydroxy alantolactone (IC_50_ = 3.2–6.4 µM). Preliminary conformational analysis indicated that oxidation of the lead compound C1-OH would show greater cytotoxicity; the double bond at the C5–C6 position might be more effective for activity. At the same time, the C1-OH esterified derivative would be less potent. Further studies revealed that compound **39** would concentration-dependently inhibit TNF-*α*-induced NF-*κ*B signaling in PC-3 cancer cells and lead to G2/M phase arrest of PC-3 cancer cells in the cell cycle [63].

Curcumin is an extremely potent natural product with numerous biological effects. Researchers improved the stability of the compounds by using heterocyclic substitution of the diketone group of curcumin. One of the isoxazole curcumin derivatives, **40** (Figure 29), exhibited potent antitumor activity, with compound **40** (IC_50_ = 3.97 µM) showing more significant cytotoxicity against the breast cancer cell line (MCF7) compared to the parent compound curcumin (IC_50_ = 21.89 µM). In addition, compound **40** consistently showed better docking fractions than the other compounds and curcumin. As indicated by preliminary conformational analyses, curcumin’s biological activity was enhanced by the introduction of the isoxazole ring, and isoxazole curcumin is promising as an anti-breast cancer drug [64].

### 2.2. Antibacterial Activity

Acridone derivatives containing isoxazoline backbone were synthesized as potential antibacterial agents by Kudryavtseva et al. Compounds **41a**, and **41b** (Figure 30) showed a high inhibitory capacity against the tested strains of the studied microorganisms, significantly exceeding the positive drug metronidazole (almost 3-fold). The in vitro antifungal activity against *B. subtilis* and *C. albicans* was determined, and compounds **41a**, and **41b** showed better antibacterial activity than furacilin and ofloxacin. Of particular note, compounds **41a**, and **41b** showed the highest activity against *C. albicans*. Antibacterial activity was enhanced by the presence of nitrofuran on the isoxazoline ring [65].

A series of sampangine derivatives containing isoxazole were synthesized by li et al. Compound **42** (Figure 31) containing isoxazole showed the strongest antibacterial activity against *C. neoformans* H99 (MIC_80_ = 0.031 µg/mL) compared to the positive control drugs voriconazole (MIC_80_ = 0.12 µg/mL) and FLC (MIC_80_ = 2 µg/mL). In addition, compound **42** showed strong fungicidal activity against resistant *C. albicans.* (MIC_80_ = 0.12 µg/mL), which was comparable to voriconazole. Subsequent further studies on the antifungal mechanism of compound **42** revealed that it could induce necrotic cytosis in *C. neoformans* cells and block the cell cycle in the G1/S phase. SAR showed that the introduction of isoxazole significantly improved the antibacterial activity of the parent compound. Adding nitro to the thiophene ring did not significantly affect compound **42**’s antibacterial activity, whereas bromine substitution produced a negative effect. A significant decrease in antimicrobial activity was observed when the quinone group was reduced or substituted with pyrone [66].

Isoxazoline derivatives of 1′-S-acetoxychavicol acetate and sclareol were prepared by 1,3-dipole cycloaddition reaction by Anuchit et al. It was determined that these compounds were antitubercular and antibacterial. Among them, 1′-S-acetoxychavicol acetate derivative **43** (Figure 32) showed the strongest activity against the *Mycobacterium tuberculosis* H37Ra strain (MIC = 17.89 µM), which was superior to its parent compound (MIC = 26.68 µM). Sclareol derivative **34b** (Figure 23) showed the strongest antitubercular activity with a MIC value of 14.58 µM. Among the derivatives of sclareol, isoxazoline derivative **34b** showed the strongest activity against *Bacillus cereus* (MIC = 29.16 µM), which was more potent than the parent compound (MIC= 162.07 µM). Thus, the derivatives with isoxazoline fraction showed more significant antibacterial and antitubercular activity than the natural parent product [58].

Imen et al. produced a number of coumarin isoxazoline derivatives and tested these compounds’ antibacterial efficacy on Gram-positive and Gram-negative bacteria in vitro. The compounds **44a**–**44d** (Figure 33) showed the best antibacterial activity out of all the compounds examined. The results showed that derivatives **44a**, **44b**, and **44c** showed significant activity against *Pseudomonas aeruginosa* (ATCC27950; MIC of 0.03 mg/mL), which was better than the control compound gentamicin (MIC = 0.5 mg/mL). Also, derivative **44a** showed better antibacterial activity against *Staphylococcus aureus, *Enterococcus faecalis*,* and *Escherichia coli* than other isoxazoline derivatives (MIC = 0.62 mg/mL). There are no substituents in benzene ring of compound **44a**, which may be why this occurs. In contrast, the para-substituted benzene derivative **44d** with NO2-substituent showed lower antibacterial activity, and the compound of the pyrrole system showed good activity against *Enterococcus faecalis* (MIC = 0.31 mg/mL) but slightly less activity than the rest of the compounds [67]. Rao et al. reported that flavonoid derivative **45** (Figure 34) containing isoxazole fraction showed moderate antifungal activity against Mycobacterium Bovis strain (BCG) with 41.7% inhibition [68].

Hispolon is a phenolic natural substance with a wide range of biological functions. Balaji et al. created a number of isoxazole and pyrazole derivatives of hispolon. Among them, isoxazole derivative **46** (Figure 35) showed the best anti-tuberculosis activity against *Mycobacterium tuberculosis* H37Rv (Mtb H37Rv) (MIC= 1.6 µg/mL). The compound exhibiting the best antituberculosis activity among pyrazole derivatives had a MIC value of 3.2 µg/mL. A comparison of isoxazole derivatives and pyrazole derivatives shows that isoxazole derivatives display a higher potency. There was a significant increase in the compound’s bioactivity when a hydroxyl group was present on the benzene ring. The hydroxyl group derivation to multiple –OCH3 or –OAc significantly decreased the bioactivity. Gram-negative and Gram-positive bacteria were used to determine the antibacterial activities of synthesized isoxazoles and pyrazoles. The isoxazole derivatives showed more significant antibacterial activity [69].

The antimycobacterial activity of curcumin isoxazole derivatives against *Mycobacterium tuberculosis* was investigated in several studies. Among them, compound **40** (Figure 29) showed the most significant inhibitory effect against *Mycobacterium tuberculosis* (MIC = 0.09 µg /mL), and its activity was vastly superior to that of the parent compound curcumin, with 18 and 2 times more activity than the positive control drugs kanamycin and isoniazid, respectively. The SAR analysis found that curcumin derivatives possess high antifungal activity because of the isoxazole ring and unsaturated bonds on the heptyl chain. The antifungal activity of the derivative was enhanced by alkoxy and hydroxyl groups on the aromatic ring. Still, complete demethylation of the isoxazole derivative resulted in a decrease in the activity of the compound [70].

In recent years, the natural product cinchonic acid was used as a precursor compound by Sahoo et al. to synthesize isoxazole ester derivatives. Among them, derivative **47a** (Figure 36) showed the most significant inhibition against Mtb H37Rv (MIC = 0.5 µg /mL). In clinical isolates of DR-Mtb, derivative **47b** (Figure 36) showed the most potent activity with MICs of 1–4 µg/mL. By structurally modifying **47b**, derivative **47ba** showed even greater potency against DR-Mtb (MIC= 0.25–0.5 µg/mL). According to SAR, compounds showed better bioactivity after substitution by halogen, the presence of alkoxy groups significantly enhanced the bioactivity, and substitution on the quinoline part decreased the bioactivity. Therefore, these compounds with isoxazolyl ester fragments are promising as effective antifungal drugs [71].

Sahoo et al. synthesized a series of isoxazole–chalcone mixtures in which compound **48** (Figure 37) showed the most significant inhibitory activity (MIC = 0.12 µg /mL) and selectivity (SI > 320) against Mtb H37Rv, which was superior to the positive control drug Streptomycin (MIC = 0.5 µg /mL). SAR studies showed that the methyl isoxazole fraction was essential for the mixture’s antitubercular activity, and the chalcone fraction enhanced the activity and selectivity of the mixture. Among them, it was found that compounds with non-polar groups, such as halogen and alkyl groups substituted on the benzene ring of chalcone, exhibited better antifungal activity. In contrast, compounds with OH groups substituted on the benzene ring significantly reduced the activity. The inhibition of Mtb H37Rv was reduced when the aryl group (R1) was substituted with a heterocyclic group. The compounds containing nitro showed good to moderate potency, despite being a polar substituent, probably related to its electronic interactions [72].

By substituting the tetrahydrofuran ring of neolignans with 1,2,3-triazole and isoxazole rings, Neves et al. synthesized novel derivatives. These derivatives were found to be active against intracellular amastigotes, with derivatives **49a**, **49b**, **49c**, and **49d** (Figure 38) exhibiting significant antileishmanial activity (IC_50_ = 0.9 µM, 0.4 µM, 0.7 µM, and 1.4 µM) showing a high selectivity index (SI = 178.0–625.0). It was observed that the isoxazole derivatives (IC_50_ = 0.4–1.4 µM) had stronger activity (IC_50_ = 4.4–29.2 µM) than the triazole derivatives with identical substituents. SAR analysis showed that trimethoxy groups were necessary for antileishmanial activity [73]. Another study showed that isoxazole derivative **50** (Figure 38) showed the most significant inhibitory activity against *L. amazonensis* and *L. braziliensis* (IC_50_ = 2.0, 1.2µM). All compounds were non-cytotoxic. Based on SAR analysis, methylenedioxy groups are essential to the antileishmanial activity [74].

### 2.3. Anti-Diabetic Activity

Algethami et al. found and discovered some isoxazole-containing derivatives of flavonoids. Among the newly synthesized compounds, compound **51** (Figure 39) was discovered to have the strongest inhibitory activity against *α*-amylase (50 µM: PI = 94.7 ± 1.2%; IC_50_ = 12.6 ± 0.2 µM), and its activity was similar to that of the positive control acarbose (50 µM: PI = 97.8 ± 0.5%; IC_50_ = 12.4 ± 0.1 µM). SAR analysis showed that the inhibition of the *α*-amylase activity of the compounds was significantly enhanced when there was a halogen atom (F, Cl, or Br) substitution at the phenyl substituent on the isoxazole ring, and the activity of the fluorinated derivative **51** was higher than that of the chlorinated and brominated derivatives. However, the compounds did not exhibit significant α-amylase inhibitory activity when there was methyl or tert-butyl substitution at the phenyl substituent, so the increase of donor-induced effect (+I) had no significant effect on the α-amylase inhibitory activity [75]. Saidi et al. created a number of isoxazoles based on halogenated flavonoids. Among them, compound **52** (Figure 39) (IC_50_ = 16.2 ± 0.3 µM) exhibited the highest anti-α-amylase activity comparable to the standard (acarbose, IC_50_ = 15.7 ± 0.2 µM). An SAR analysis revealed that bromine atoms are essential for α-amylase inhibitory activity [76].

A group of spiroisoxazoline-containing glucose derivatives were created by Goyard et al., and their potential anti-diabetic properties were examined. Among them, compounds **53a** and **53b** (Figure 40) were found to be the most potent inhibitors of rabbit muscle glycogen phosphorylase b with an IC_50_ value of 1.54 µM. Further studies revealed that compounds **53a** and **53b** inhibited targeted glycogenolysis via GP in cellular models. According to the SAR analysis, compounds containing the 2-naphthyl fraction had better inhibitory potency, and changing the 6-OH substituent of the naphthyl group in compound **53b** group to methoxy (**53c**) in compound **53b** resulted in weaker inhibitory potency, suggesting that 6-OH may be involved in favorable hydrogen bond formation [77].

Stilbene scaffolds can be found in a wide range of biologically active natural products. Some laboratories have recently synthesized a variety of isoxazole-containing stilbene derivatives; A colorimetric assay was used to determine if these substances might inhibit the activity of protein tyrosine phosphatase 1B (PTP1B) and TCPTP. Among them, compound **54a** (Figure 41) showed the best inhibitory activity IC_50_ values of 0.91 ± 0.33 µM and 5.19 ± 0.31 µM, respectively, which was more potent than the lead compound lithocholic acid (IC_50_ = 12.54 ± 2.51 µM). Compound **54b** (Figure 41) showed significant activity and the best selectivity (TCPTP/PTP1B = 20.7). Based on SAR analysis, compounds containing chlorine or dichlorine substitutions showed the greatest inhibitory activity against enzymes [78].

Arjunolic acid can be isolated from natural plants and has a wide range of biological activities. The research team synthesized a new phenylethynyl and isoxazole derivatives based on the arjunolic acid structure. It was found that compounds **55a** and **55b** (Figure 42) exhibited the best inhibitory activity against tyrosinase and α-glucosidase. Among them, compound **55a** showed the strongest tyrosinase inhibition (IC_50_ = 14.3 ± 7.6 µM), superior to the positive control drug kojic acid (IC_50_ = 41.5 ± 1.0 µM). Furthermore, compound **55b** (IC_50_ = 14.5 ± 0.15 µM) exhibited α-glucosidase inhibition comparable to the standard drug acarbose (IC_50_ = 10.4 ± 0.06 µM). Thus arjunolic acid derivatives **55a** and **55b** containing isoxazole showed stronger activity than the parent compounds and are potential anti-diabetichouxuan drugs [79].

Kaempferol is a naturally occurring flavonoid compound that shows potent anti-diabetic activity. Nie et al. synthesized a series of flavonoid derivatives containing triazole or isoxazole by attaching triazole or isoxazole rings to C7-OH by carbon chains of different lengths. Among them, the isoxazole derivatives containing 1-carbon spacers **56a–56c** (Figure 43) showed the most significant improvement in glucose depletion in IR HepG2 cells (EC_50_ = 0.8–2.9 µM), and their activity was superior to that of the isoxazole derivatives with 2-carbon spacers (EC_50_ = 46.0–89.0 µM). And the isoxazole derivatives exhibited better biological activity than the triazole derivatives. Follow-up studies suggest that the potential molecular mechanism of isoxazole derivatives **56a–56c** may activate the AMPK/PEPCK/G6Pase pathway. Thus, Kaempfero’s novel isoxazole derivative may be a promising anti-diabetic drug candidate [80].

### 2.4. Anti-Inflammatory Activity

After synthesizing novel glucocorticoid isoxazoline derivatives, researchers tested them for anti-inflammatory efficacy in vitro. Among all compounds screened, compound **57** (Figure 44) showed promising NO and IL-8 inhibitory activity. Among Raw264.7 mouse macrophages, **57** dose-dependently inhibited LPS-induced NO release with 10-fold higher potency (IC_50_ = 6 nM) than dexamethasone. In human airway smooth muscle cells, **57** concentration-dependently inhibited TNF-α-induced IL-8 release with a potency (IC_50_ = 0.84 nM) comparable to dexamethasone. Preliminary SAR analysis indicated that introducing bromine substituents in isoxazoline derivatives enhanced the anti-inflammatory activity of the compounds [81].

The natural alkaloid sinomenine, which can be extracted from plant roots, has a variety of biological properties. Sinomenine derivatives containing the isoxazoline fraction were synthesized and tested in vitro for anti-inflammatory properties by Jin et al. The inhibitory activity of sinomenine derivatives against TNF-*α*-induced NF-*k*B activation was studied at a concentration of 20 µM. It was seen that isoxazoline derivatives **58a** and **58b** (Figure 45) improved the anti-inflammatory activity to some extent. However, the compounds introduced with cinnamic acid esters exhibited stronger inhibitory activity than the isoxazoline derivative [82]. Due to the excellent activity of sinomenine isoxazolines, this team synthesized a series of novel sinomenine isoxazole derivatives in 2019, with the highest-yielding compound being **59** (Figure 45). In the future, these compounds promise to develop into new anti-inflammatory drugs [83].

A series of novel isoxazole and pyrazole derivatives were synthesized based on the natural product biphenyl-neolignans honokiol by Yuan et al. and evaluated their in vitro anti-inflammatory activities. Inactivated BV-2 microglia, isoxazole derivatives **60a** and **60b** (Figure 46) exhibited moderate inhibitory activity against NOS-mediated nitric oxide production with IC_50_ values of 25.9 and 28.7 µM, respectively. According to SAR analysis, isoxazole derivatives inhibit NO production by attaching an allyl substituent to the C5 position of the benzene ring. And this study found that pyrazole derivatives showed more potent inhibitory activity than isoxazole derivatives and parent compounds [84].

The natural product karanja is found in the seeds of Indian medicinal trees and has a wide range of biological properties. Rekha et al. structurally modified karanja to obtain an isoxazole derivative **61** (Figure 47). Mouse-ear swelling models induced by xylene were used to evaluate the derivative’s anti-inflammatory activity in vivo. Compound **61** showed more active anti-inflammatory activity (inhibition, 75.45%) than the parent compound karanja (51.13% inhibition) and was comparable to the standard drug ibuprofen (77.27% inhibition). It can be seen that the isoxazole derivatives significantly reduced the effect of inflammatory mediators, thus inhibiting ear edema more effectively than its lead compound [85].

A variety of isoxazole, pyrazole, and pyrimidine derivatives of curcumin were synthesized by Ahmed et al. Next, they used a mouse model of carrageenan-induced paw edema to assess the derivatives’ anti-inflammatory efficacy in vivo. Among them, isoxazole derivative **40** (Figure 29) exhibited relatively active anti-inflammatory activity (inhibition rate, 66.1%) over the parent compound curcumin (45.7% inhibition rate). The inhibitory activity of all heterocyclic derivatives against human COX-2 enzyme was also evaluated at a concentration of 10 µM, and it was found that isoxazole derivatives exhibited 49.3% inhibition, which was lower than pyrimidine derivatives (inhibition, 75.3%) and pyrazole derivatives (inhibition, 55.9%). The final results showed that the anti-inflammatory activity of the pyrimidine curcumin derivatives was stronger than that of the isoxazole and pyrazole derivatives [86].

### 2.5. Insecticidal Activity

Natural bicyclic sesquiterpene (+)-nootkatone is derived from Alaska yellow cedar, *Citrus Rutaceae.* A group of derivatives of (+)-Nootkatone that contain the isoxazoline portion were created and synthesized by Guo et al. Derivatives **62a** and **62b** (Figure 48) displayed the strongest insecticidal efficacy of all of them. Compounds **62a** and **62b** exhibited better growth inhibition activity against Mucor, the final mortality rates (FMRs) both 73.3%, higher than the positive control toosendanin (50.0%) and 1.7 times higher than nootkatone (43.3%). In addition, derivative **62c** (Figure 48) exhibited significant larvicidal activity with an LC_50_ value of 0.23 µmol mL^−1^. The analysis of SAR showed that the substitution of the aromatic ring affected the biological activity. The halogen and powerful electron-absorbing groups added to the benzene ring boosted Mucor’s insecticidal effectiveness: dihalogenated groups > monohalogenated groups; electron-absorbing groups (F, Cl, Br, NO_2_) > electron-giving groups (OMe, Me) [87].

Podophyllotoxin is a natural aromatic lignan extracted from roots and rhizomes of *Podophyllum hexandrum*. Yang et al. synthesized and evaluated a series of isoxazole-containing podophyllotoxin derivatives for their insecticidal activity. The researchers found that compounds **63a** and **63b** (Figure 49) exhibited the most potent insecticidal activity against Mucor and Vibrio mites. Compounds IIIc and IIId showed better growth inhibitory activity against Mucor mites compared to the positive control toosendanin, The final mortality rate was 69.0% and 62.1%, respectively. Moreover, podophyllotoxin showed almost no acaricidal activity against *C. vermicularis*, whereas its isoxazole derivatives showed more effective acaricidal action than the lead compounds. Compounds IIIc and IIId showed MRS of 41.1% and 32.8% at 72 h. SAR studies showed that introducing a chlorine atom at the C-2′ position and a chlorine/fluorine atom at the C-4‘ position of the isoxazole fragment of podophyllotoxin enhances the acaricidal activity of the compounds [88].

Xu et al. synthesized a series of cholesterol derivatives containing isoxazoline/isoxazole fragments. They found that compounds **64a**, **64b**, and **65a** (Figure 50) showed better growth inhibition activity against *Aphis citricola*; the corrected mortality rate was 70.3%, 62.9%, and 66.6%, respectively. In addition, compounds **65b** (Figure 50) showed 5.8 times more insecticidal activity than cholesterol against *Aphis citricola*. SAR studies showed that cholesterol modification at both C-3 hydroxyl and C-7 sites significantly increased the insecticidal activity of the compounds. Interestingly, the insecticidal activity of isoxazoline compounds against *Plutella xylostella* was more significant than the corresponding isoxazoline compounds [89].

According to a study from Liu et al., the ostiole-based isoxazoline derivatives that were designed and synthesized exhibit a wide range of biological activity. Among them, derivative **66a** (Figure 51) showed better growth inhibitory activity against *Cnidium monnieri* with a CMR of 96.4% at 30 days, which was higher than the positive control agent toosendanin (53.6%) and 1.80 times higher than that of ostiole (53.6%). Meanwhile, derivative **66b** (Figure 51) exhibited significant larvicidal activity against the small cabbage moth with an LC_50_ value of 0.22 mg/mL, which was superior to rotenone (LC_50_ = 0.41 mg/mL). According to SAR studies, introducing halogen atoms to benzene rings attached to isoxazoline improved insecticidal activity against stick insects and larvae. The results suggest that osthole derivatives containing isoxazoline can be further investigated as natural insecticides [90]. In another study, a series of ((2′E)-4′-(isoxazolin-5″-yl)carbonyloxyosthole derivatives containing isoxazolin fragments were synthesized. Among them, compound **67a** (Figure 51) showed the strongest vermilion mite-killing activity (LC_50_ = 0.76 mg/mL), which was superior to the lead compound ostiole (LC_50_ = 1.14 mg/mL). For stick insects, compounds **67b** (FMR:55.1%) and **67c** (FMR:62.0%) (Figure 51) showed better growth inhibitory activity at 1 mg/mL, which was 1.5–1.6 times higher than that of ostiole (37.9%). Several SAR studies revealed that stick and mite insecticidal activity could be enhanced by adding acryloyloxy-linked isoxazoline in the parent compound [91].

### 2.6. Other Biological Activity

A novel isoxazole chalcone derivative **68** (Figure 52) was synthesized, and its biological activity was evaluated by Li et al. The results showed that compound **68** (50 µM, 135.7 ± 9.0%) exhibited stronger activity against tyrosinase in mouse B16 melanoma cells compared to the positive control drug 8-MOP (50 µM, 120.1 ± 2.9%). Also, compound **68** was found to be effective in promoting melanin synthesis in B16 cells, and its activity (50 µM, 199.8 ± 18.1%) was superior to that of the positive control drug 8-MOP (50 µM, 127.9 ± 18.5%). A Western blotting assay showed that compound **68** promotes melanogenesis through Akt and GSK3β signaling pathways, which is promising as a potential therapeutic agent for vitiligo in the future [92].

Xian et al. designed and synthesized coumarin derivatives containing the isoxazole fraction and evaluated their biological activities. Compounds **69a** and **69b** (Figure 53) were found to be effective in promoting melanin synthesis in murine B16 melanoma cells. Compounds **69a** (242%) and **69b** (390%) showed better activity than the positive control drug 8-MOP (149%). SAR studies showed that the number of halogen atoms on the benzene ring significantly affected the activity. The introduction of two Cl atoms into benzene significantly increased the activity of the compounds, and the 3,5-disubstituted was more active than the 3,4-disubstituted compounds, thus compounds **69a** and **69b** containing isoxazole modifications could be used as potential anti-papillary drug candidates [93].

It was reported by Wu et al. that OA derivatives containing isoxazole and pyrazole fractions were synthesized. A series of synthesized compounds were tested for their ability to inhibit RANKL-induced osteoclast differentiation from RAW264.7 cells. The results showed that pyrazole derivative **70** (Figure 54) (90.0% inhibition) showed better inhibitory activity than isoxazole derivative **71** (Figure 54) (78.5% inhibition) and OA (11.4% inhibition). Therefore, pyrazole derivatives could be a promising anti-osteoporosis drug candidate [94].

The researchers synthesized a series of derivatives containing pyrazole, pyridopyrazotriazine, isoxazoline, and pyridine using curcumin as a backbone. They were also evaluated for their biological activities. The antioxidant activity of the newly synthesized compounds was studied using the ABTS method. The results showed compound **72** (Figure 55), containing the isoxazoline fraction, showed higher antioxidant capacity than ascorbic acid and other compounds. Also, compound **72** exhibited high protection against DNA damage induced by bleomycin–iron complexes. SAR studies showed that the presence of isoxazoline and triazine fractions showed better antioxidant capacity. The lead compound curcumin showed better antioxidant activity than the newly synthesized compounds [95]. Another study found that curcumin isoxazole derivative **73** (Figure 55) exhibited the best antioxidant activity. The percentage inhibition value of compound **73** against DPPH was greater than that of curcumin and other azoles. Moreover, in the DPPH bioassay, the EC_50_ value of isoxazole derivative **73** was 8 ± 0.11 µM, which was more active than pyrazole derivatives (EC_50_ = 14 ± 0.18 µM) and curcumin (EC_50_ = 40 ± 0.06 µM) [96].

Ahmed et al. synthesized sulfonamides with a curcumin scaffold, of which compound **74** (Figure 55) containing isoxazole showed the highest inhibitory activity against carbonic anhydrase isoenzyme I (human) with an IC_50_ value of 2.11µM. In addition, compounds containing isoxazole, pyrazole, and dihydropyrimidine (IC_50_ values of 0.97, 0.58, and 0.88 µM, respectively) showed better inhibitory activity against bCAI compared to the positive drug acetazolamide (IC_50_ = 0.94 µM). The compounds containing isoxazole, pyrazole, and dihydropyrimidine (IC_50_ values of 0.97, 0.58, and 0.88 µM, respectively) showed better inhibitory activity against bCAI. Studies using SAR showed that curcumin isoxazole sulfonamides containing acetamide at the terminal position significantly enhanced the compound’s activity [97].

Minassi et al. obtained hydroxamates isoxazole derivatives by modification of betulinic acids. The effect of **75** (Figure 56) on HIF-1*α* expression was evaluated under normal and hypoxic conditions and was found to show good biological activity (EC_50_ = 2.4 µM) as a novel HIF prolyl hydrolase inhibitor. It was observed that the hydroxamic acid fraction is essential for the activity of the derivatives, and the compound loses its activity when the hydroxamic acid hydroxyl group is alkylated. At the same time, the introduction of isoxazole significantly enhances the biological activity of the compound [98].

Qiu et al. synthesized a series of novel isoxazole-chenodeoxycholic acid mixtures and evaluated the lipid-lowering effect of all their mixtures using the 3T3-L1 adipocyte model. Compound **76** (Figure 57), containing an N-methyl amide group, was found to have a significant lipid-lowering effect, reducing lipid accumulation by 30.5% at 10 µM. According to SAR, isoxazole heterocycles would enhance compound activity. When the amide groups of the compounds were attached to groups with excellent spatial site resistance, their hypolipidemic activity was reduced. Further studies on compound **76** revealed that **76** inhibited lipid accumulation in 3T3-L1 adipocytes via the FXR-SHP-SRBP1c signaling pathway [99].

An evaluation of the cardiovascular activity of heterocycle derivatives of panaxatriol was carried out by researchers. In biological studies, all isoxazole ring-conjugated panaxatriol derivatives showed greater cell viability than the parent compounds, with compound **77** (Figure 58) increasing cell viability to approximately 89% at the highest concentration (10 µM). According to in vivo studies, treatment with positive control captopril (20 mg/kg) reduced the myocardial infarct area by 34.0%, and treatment with panax ginseng powder (40 mg/kg) reduced the myocardial infarct area by 30.6%. However, compound **77** reduced the myocardial infarct area to 20.5% in rats. Compared to panaxatriol, isoxazole panaxatriol derivative **77** exhibited better cardiac cytoprotective effects. In SAR studies, the isoxazole ring was introduced into panaxatriol to significantly increase its cytoprotective effects, and the hydrolysis of the methoxycarbonyl group in the compound to the carboxyl group led to a significant increase in cytoprotective activity [100].

## 3. Conclusions and Perspectives

After an in-depth study of the literature, it was found that isoxazoles and isoxazolines are a class of scaffolds with a wide range of pharmacological properties in medicinal chemistry. Notably, since they are associated with a wide range of activities such as anti-inflammatory, anti-cancer, antibacterial, and anti-parasitic, synthetic modification of this pharmacophore will likely enhance the biological properties of the natural products, resulting in potent new molecules. Therefore their use for improving the activity of natural drugs is discussed in detail in this review.

Natural and synthetic isoxazoles and isoxazolines have a wide range of therapeutic applications. Researchers commonly use 1,3-dipole cycloaddition reactions on natural products to generate derivatives containing isoxazoles or isoxazolines. The literature suggests that natural products modified by isoxazole/isoxazoline are most effective against cancer diseases, targeting a wide range of cancer cell lines. Isoxazoles and isoxazolines modify the same natural product, sometimes resulting in different biological activities. In order to find natural product isoxazole/isoxazoline hybrids that may be very active biologically, we examine the research progress made over the previous 10 years in the synthesis of isoxazole/isoxazoline derivatives of natural products. In conclusion, isoxazole/isoxazoline radicals play an important role in the synthesis of many drugs and have drawn considerable interest from researchers all over the world.

## Figures and Tables

**Figure 1 pharmaceuticals-16-00228-f001:**
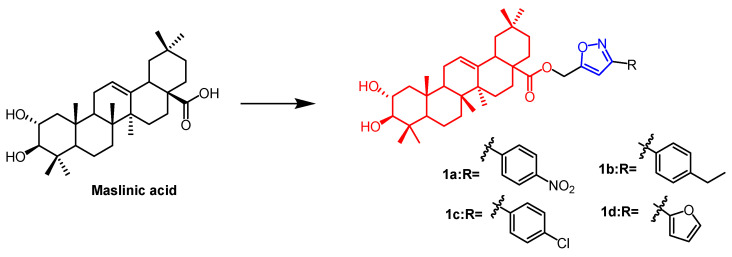
The chemical structure and derivative of maslinic. (The red marker in the figure indicates the parent structure of MA, and the blue marker indicates the structural modification of isoxazole.) [26].

**Figure 2 pharmaceuticals-16-00228-f002:**
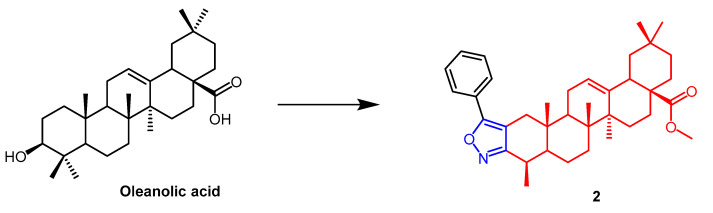
The chemical structure and derivatives of oleanolic. (The red marker in the figure indicates the parent structure of OA, and the blue marker indicates the structural modification of isoxazole.) The same explanation for the following figures [27].

**Figure 3 pharmaceuticals-16-00228-f003:**
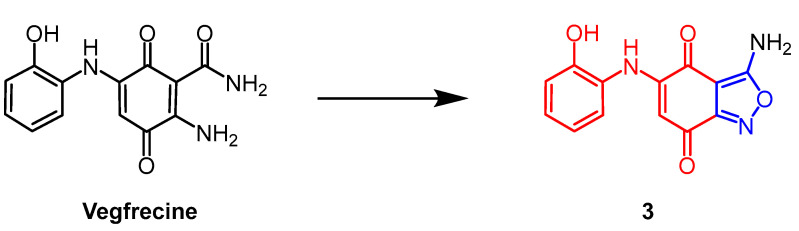
The chemical structure and derivatives of vegfrecine [28].

**Figure 4 pharmaceuticals-16-00228-f004:**
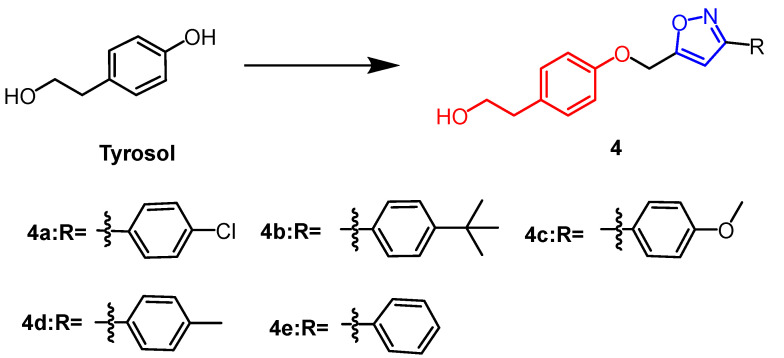
The chemical structure and derivatives of tyrosol [29].

**Figure 5 pharmaceuticals-16-00228-f005:**
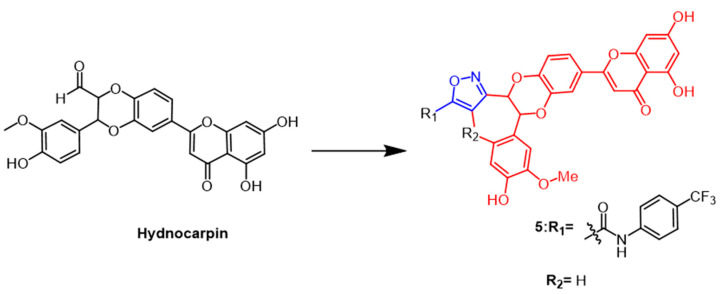
The chemical structure and derivatives of hydnocarpin [31].

**Figure 6 pharmaceuticals-16-00228-f006:**
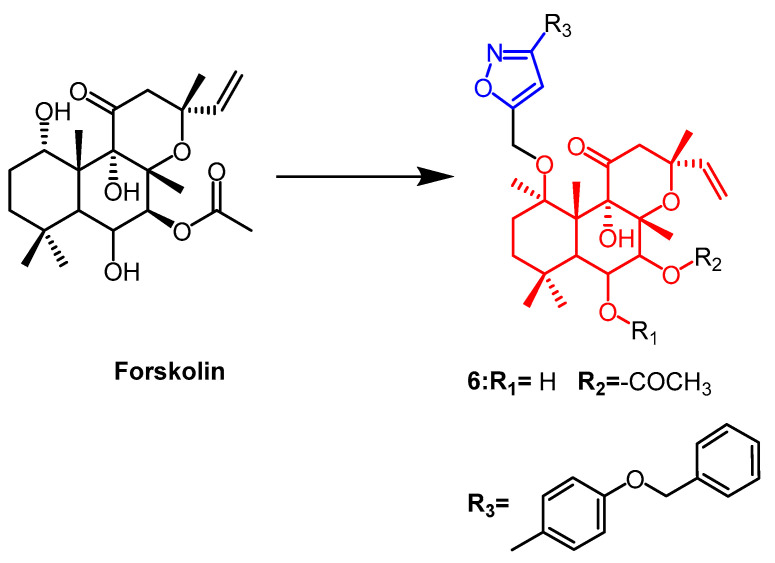
The chemical structure and derivatives of forskolin [32].

**Figure 7 pharmaceuticals-16-00228-f007:**
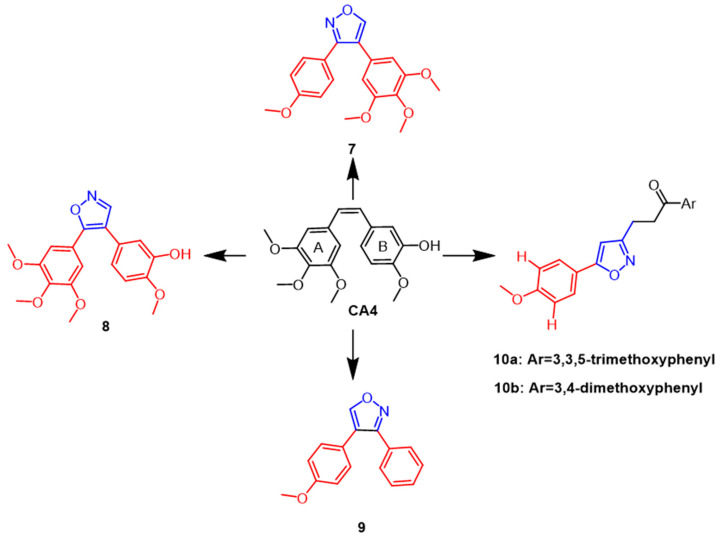
The chemical structure and derivatives of CA4 [35,36].

**Figure 8 pharmaceuticals-16-00228-f008:**
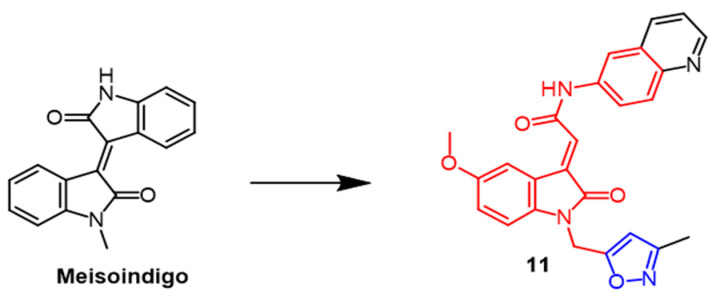
The chemical structure and derivatives of meisoindigo [37].

**Figure 9 pharmaceuticals-16-00228-f009:**
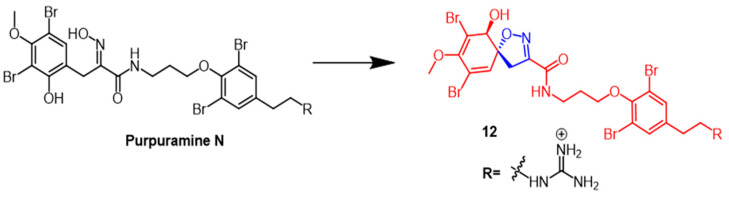
The chemical structure and derivatives of pupuramine N [38].

**Figure 10 pharmaceuticals-16-00228-f010:**
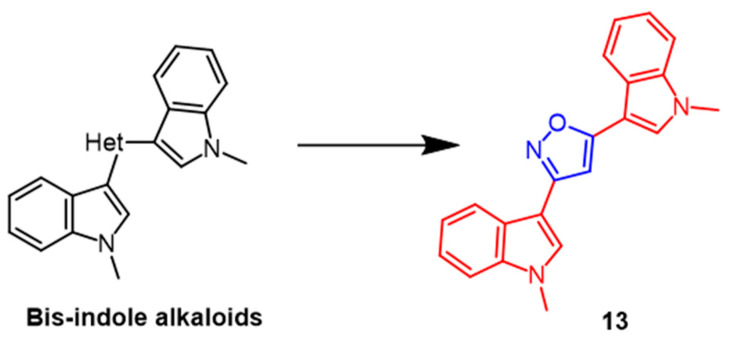
The chemical structure and derivatives of *bis*-indole alkaloids [39].

**Figure 11 pharmaceuticals-16-00228-f011:**
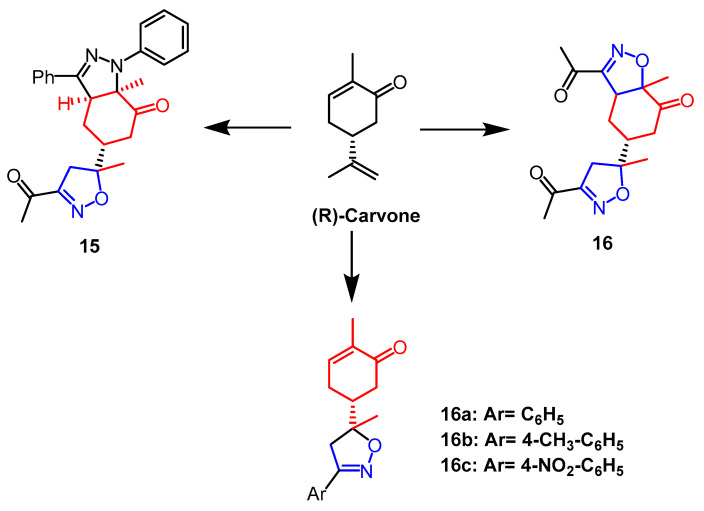
The chemical structure and derivatives of (R) -carvone [40,41].

**Figure 12 pharmaceuticals-16-00228-f012:**
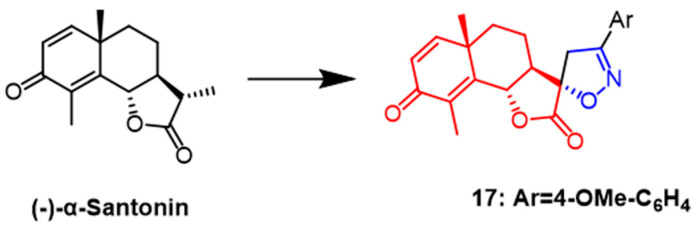
The chemical structure and derivatives of (–)-*α*-santonin [43].

**Figure 13 pharmaceuticals-16-00228-f013:**
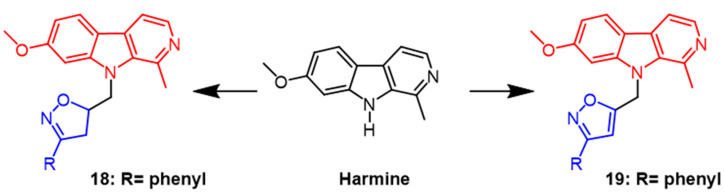
The chemical structure and derivatives of phenyl [45,46].

**Figure 14 pharmaceuticals-16-00228-f014:**
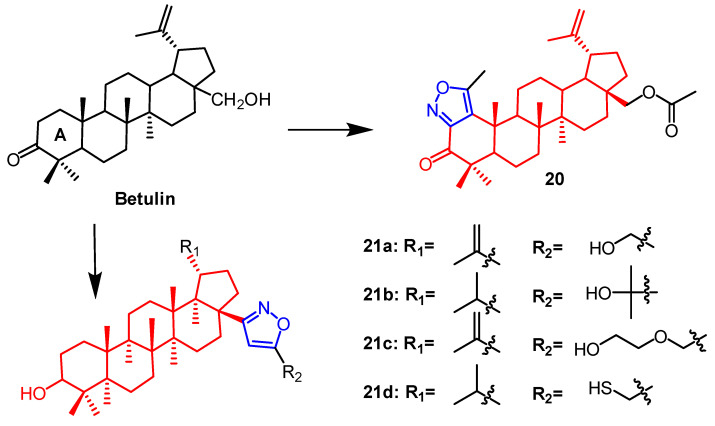
The chemical structure and derivatives of betulin [47].

**Figure 15 pharmaceuticals-16-00228-f015:**
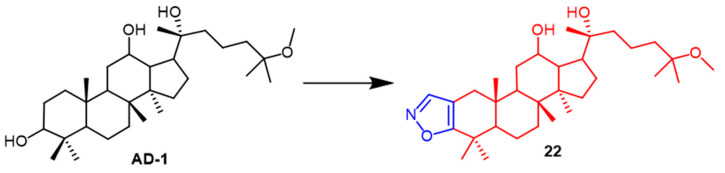
The chemical structure and derivatives of AD-1 [48].

**Figure 16 pharmaceuticals-16-00228-f016:**
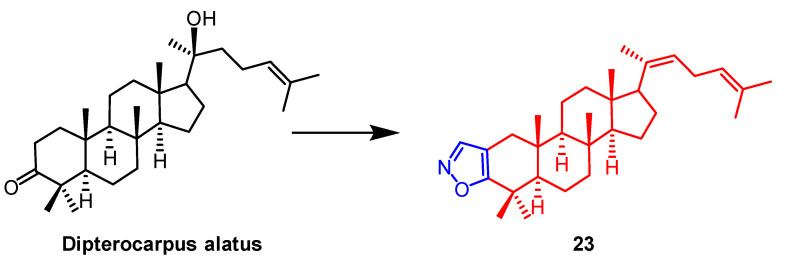
The chemical structure and derivatives of *Dispterocarous alatus* [49].

**Figure 17 pharmaceuticals-16-00228-f017:**
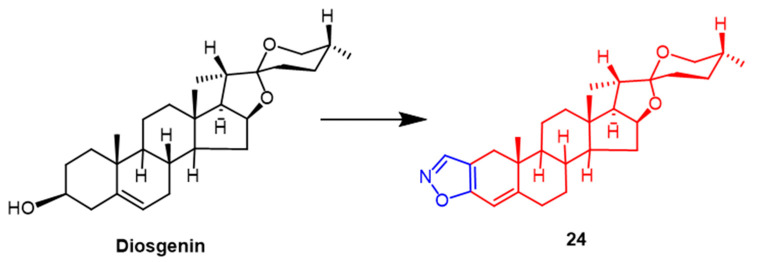
The chemical structure and derivatives of diosgenin [50].

**Figure 18 pharmaceuticals-16-00228-f018:**
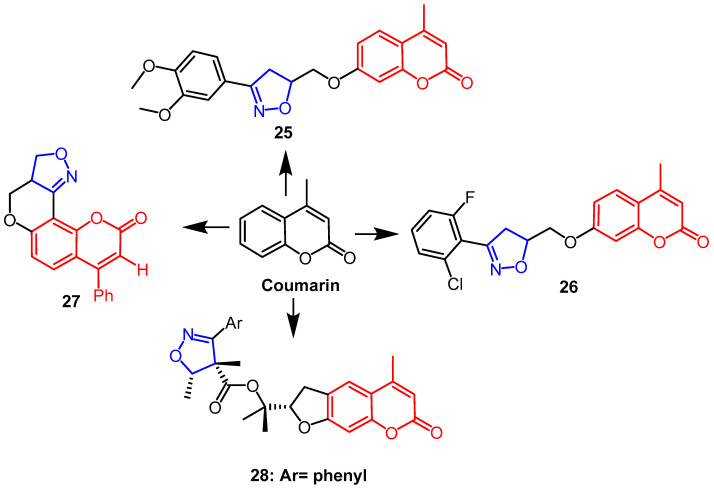
The chemical structure and derivatives of coumarin [51,52,53].

**Figure 19 pharmaceuticals-16-00228-f019:**
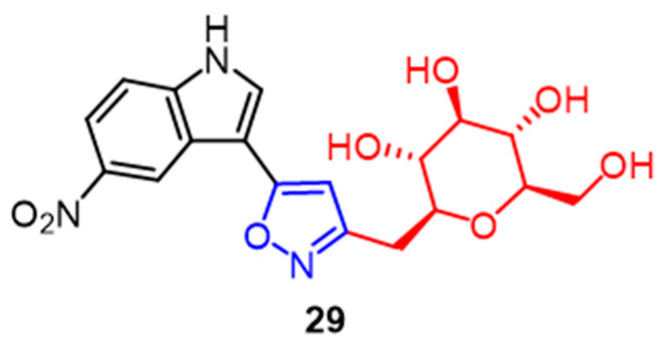
The chemical structure and derivatives of c-glycosides [54].

**Figure 20 pharmaceuticals-16-00228-f020:**
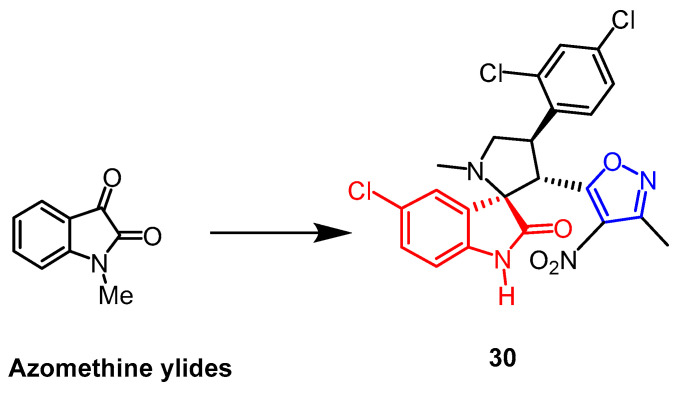
The chemical structure and derivatives of azomethine ylides [55].

**Figure 21 pharmaceuticals-16-00228-f021:**
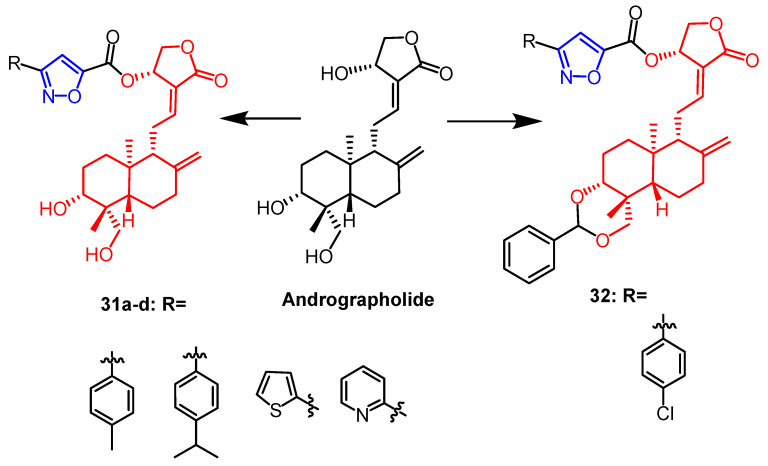
The chemical structure and derivatives of andrographolide [56].

**Figure 22 pharmaceuticals-16-00228-f022:**
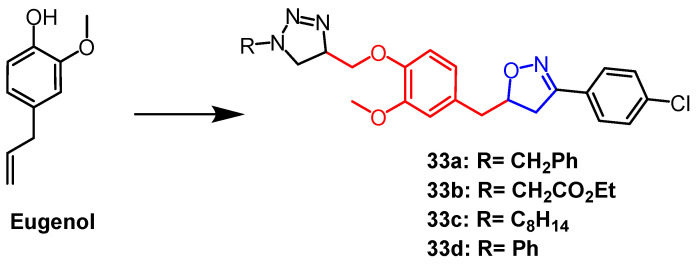
The chemical structure and derivatives of eugenol [57].

**Figure 23 pharmaceuticals-16-00228-f023:**
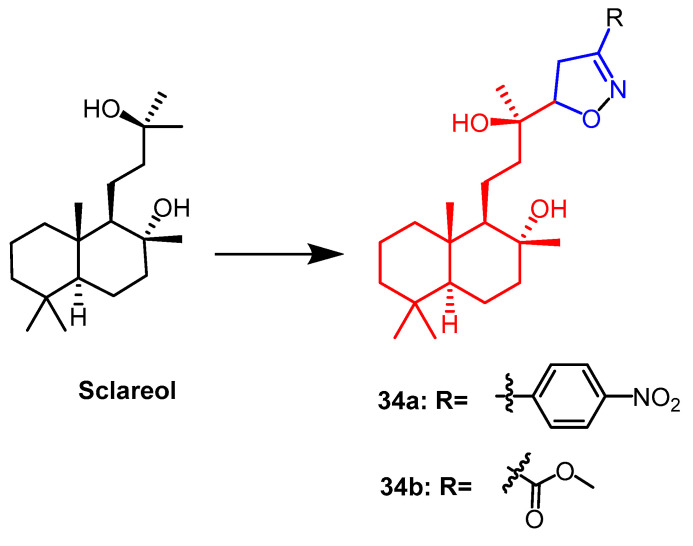
The chemical structure and derivatives of sclareol [58].

**Figure 24 pharmaceuticals-16-00228-f024:**
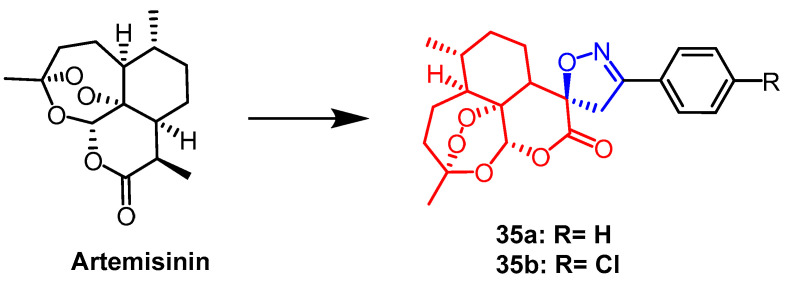
The chemical structure and derivatives of artemisinin [59].

**Figure 25 pharmaceuticals-16-00228-f025:**
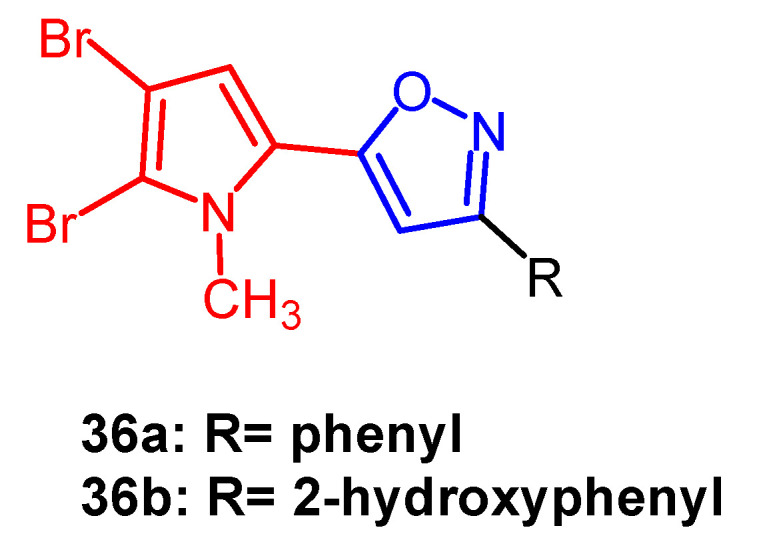
The chemical structure and derivatives of bromopyrrole [60].

**Figure 26 pharmaceuticals-16-00228-f026:**
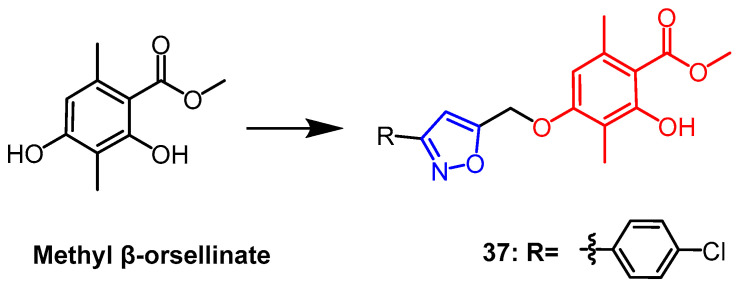
The chemical structure and derivatives of methyl *β*-orsellinate [61].

**Figure 27 pharmaceuticals-16-00228-f027:**
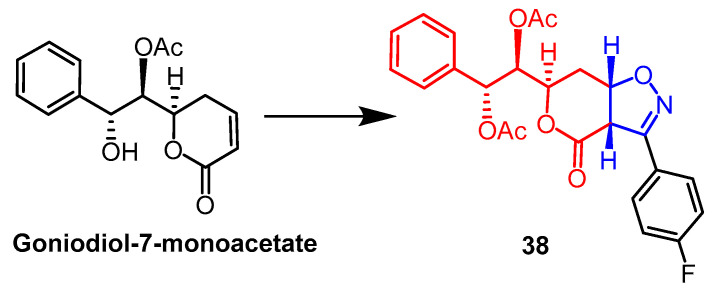
The chemical structure and derivatives of goniodiol-7-monoacetate [62].

**Figure 28 pharmaceuticals-16-00228-f028:**
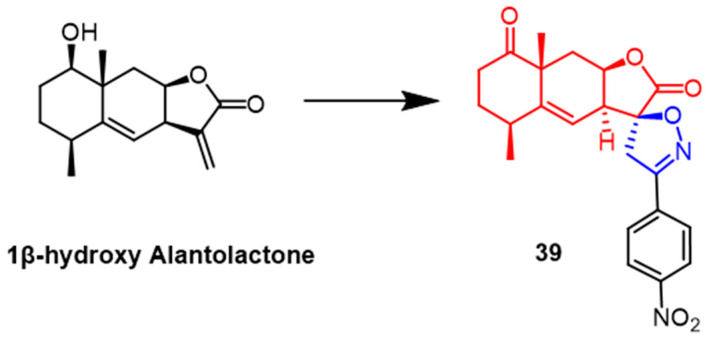
The chemical structure and derivatives of 1*β*-hydroxy alantolactone [63].

**Figure 29 pharmaceuticals-16-00228-f029:**
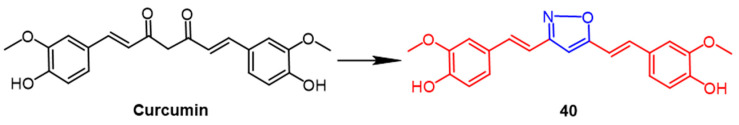
The chemical structure and derivatives of curcumin [64].

**Figure 30 pharmaceuticals-16-00228-f030:**
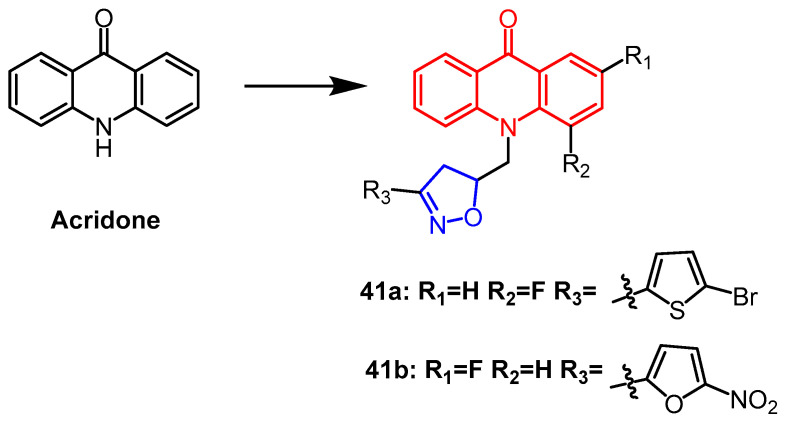
The chemical structure and derivatives of acridone [65].

**Figure 31 pharmaceuticals-16-00228-f031:**
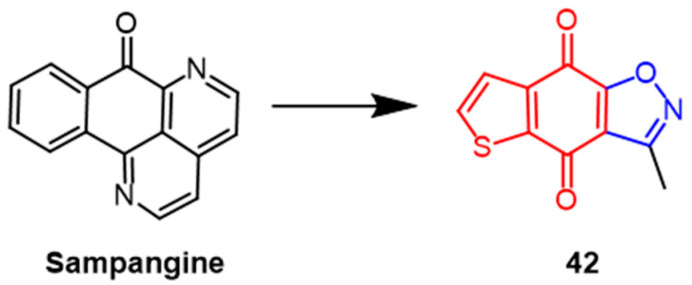
The chemical structure and derivatives of methyl sampangine [66].

**Figure 32 pharmaceuticals-16-00228-f032:**
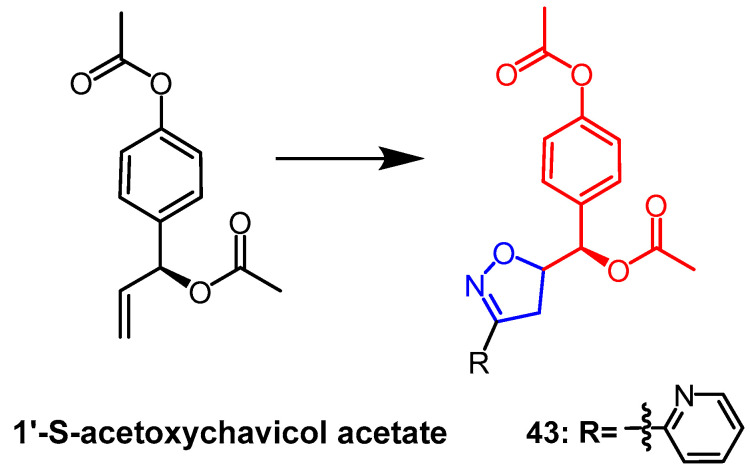
The chemical structure and derivatives of 1′-S-acetoxychavicol acetate [58].

**Figure 33 pharmaceuticals-16-00228-f033:**
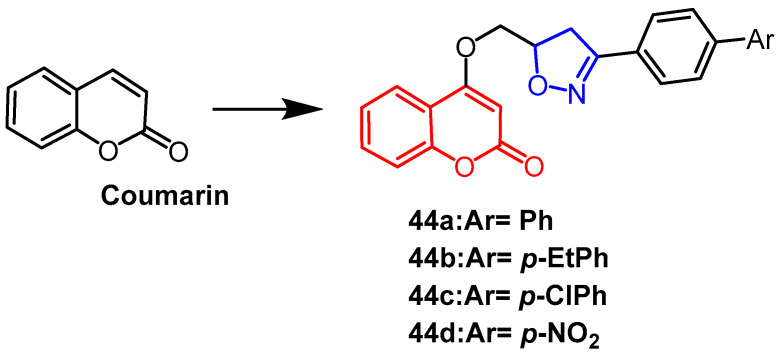
The chemical structure and derivatives of coumarin [67].

**Figure 34 pharmaceuticals-16-00228-f034:**
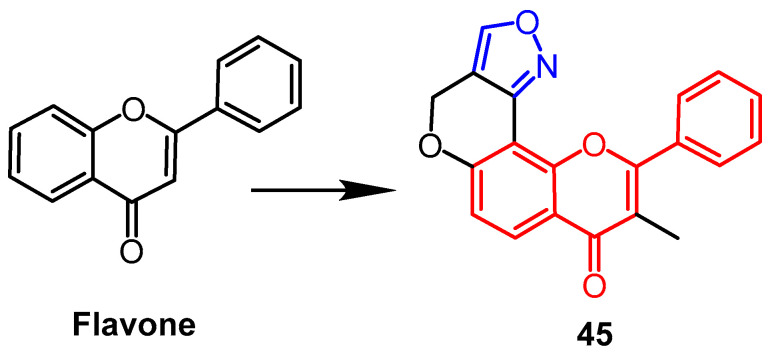
The chemical structure and derivatives of flavone [68].

**Figure 35 pharmaceuticals-16-00228-f035:**
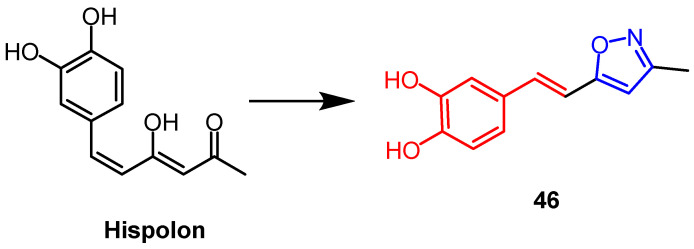
The chemical structure and derivatives of hispolon [69].

**Figure 36 pharmaceuticals-16-00228-f036:**
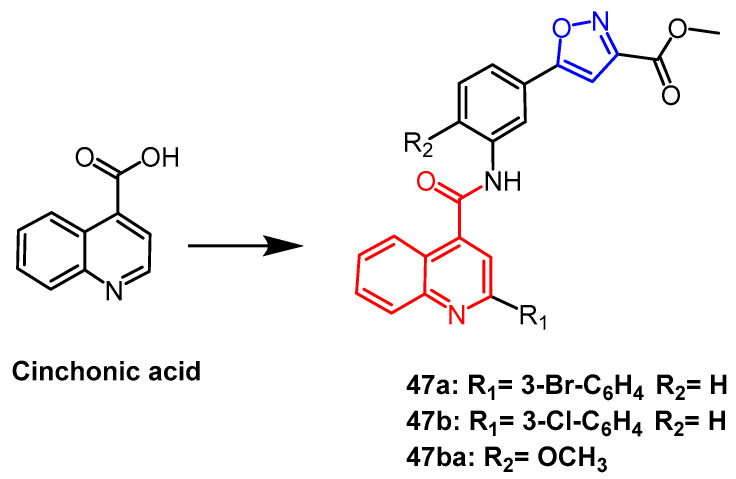
The chemical structure and derivatives of cinchonic acid [71].

**Figure 37 pharmaceuticals-16-00228-f037:**
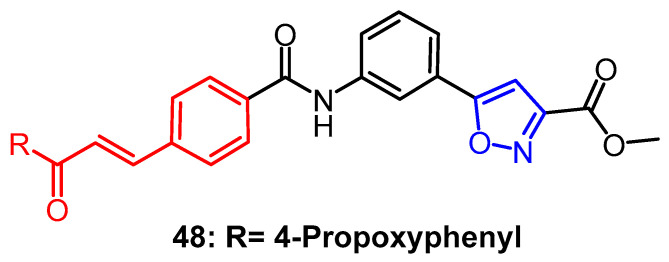
The derivatives of chalcone [72].

**Figure 38 pharmaceuticals-16-00228-f038:**
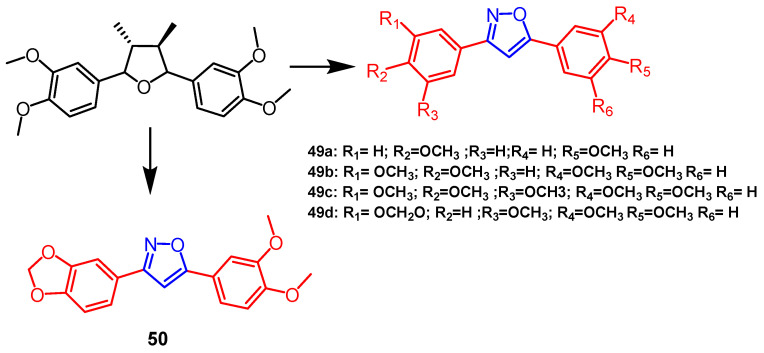
The chemical structure and derivatives of neolignans [73,74].

**Figure 39 pharmaceuticals-16-00228-f039:**
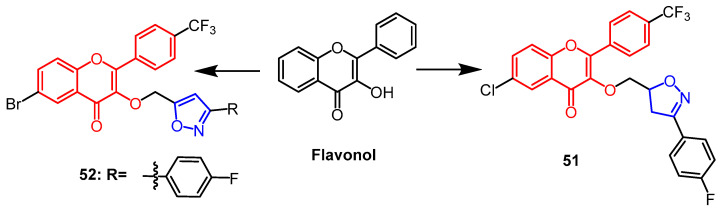
The chemical structure and derivatives of flavonol [75,76].

**Figure 40 pharmaceuticals-16-00228-f040:**
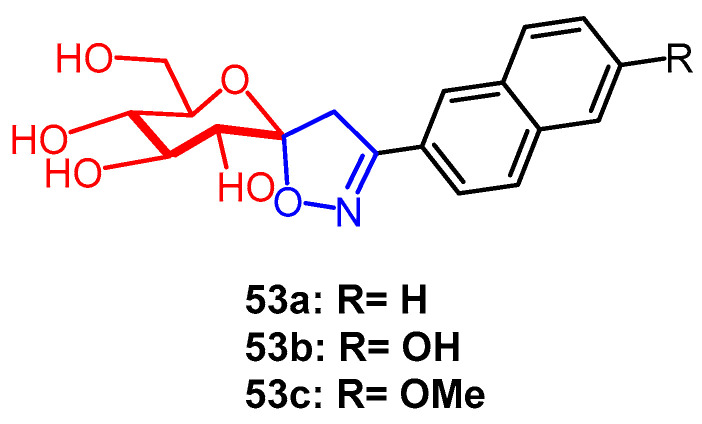
The derivatives of glucose [77].

**Figure 41 pharmaceuticals-16-00228-f041:**
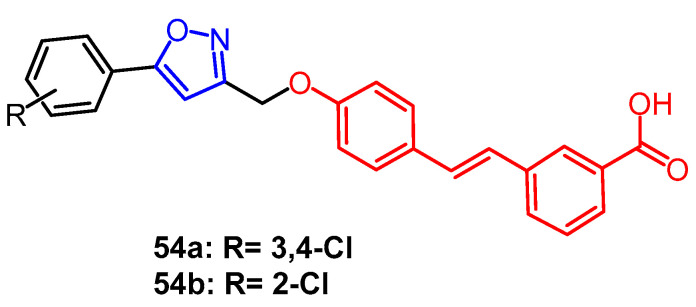
The derivatives of stilbene [78].

**Figure 42 pharmaceuticals-16-00228-f042:**
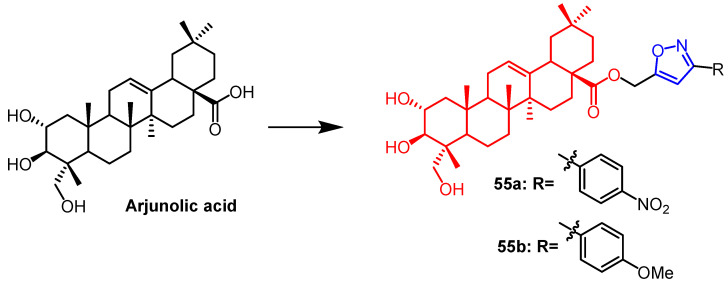
The chemical structure and derivatives of arjunolic acid [79].

**Figure 43 pharmaceuticals-16-00228-f043:**
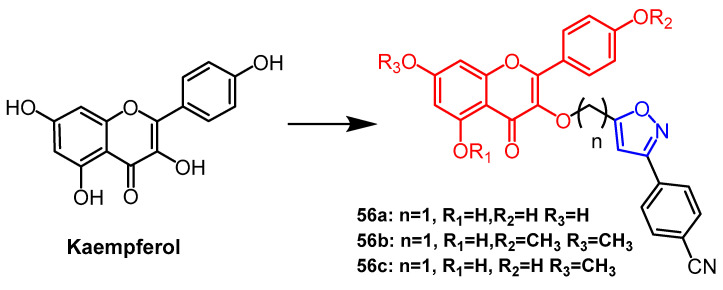
The chemical structure and derivatives of kaempferol [80].

**Figure 44 pharmaceuticals-16-00228-f044:**
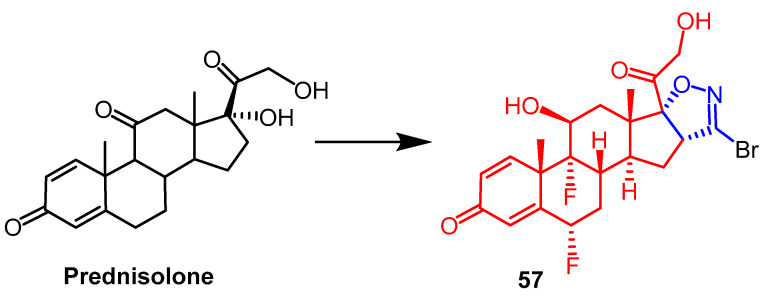
The chemical structure and derivatives of prednisolone [81].

**Figure 45 pharmaceuticals-16-00228-f045:**
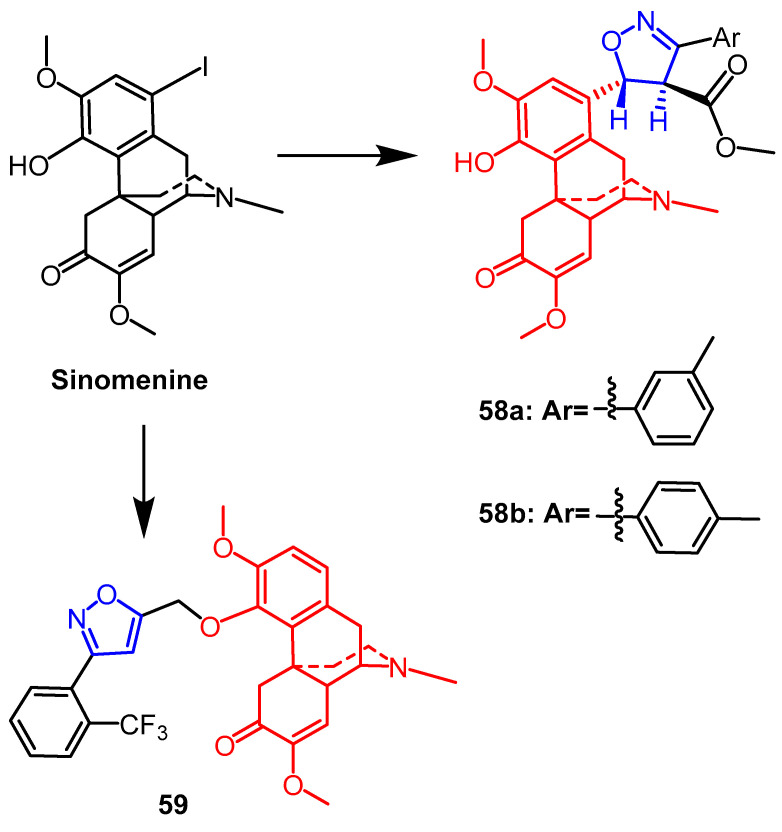
The chemical structure and derivatives of sinomenine [82,83].

**Figure 46 pharmaceuticals-16-00228-f046:**
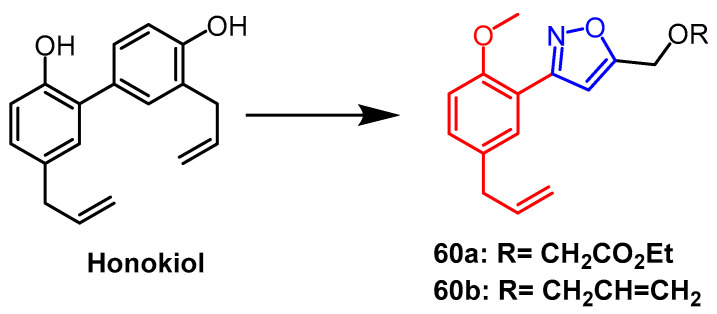
The chemical structure and derivatives of honokiol [84].

**Figure 47 pharmaceuticals-16-00228-f047:**
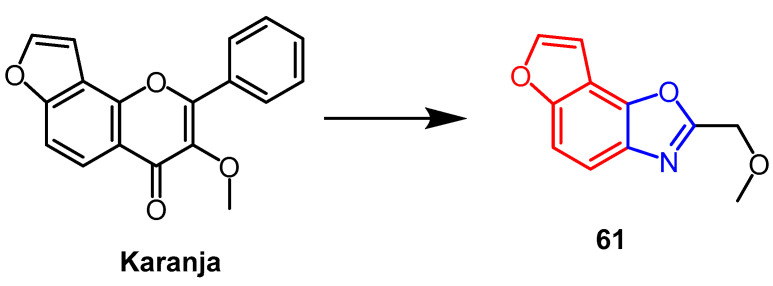
The chemical structure and derivatives of karanja [85].

**Figure 48 pharmaceuticals-16-00228-f048:**
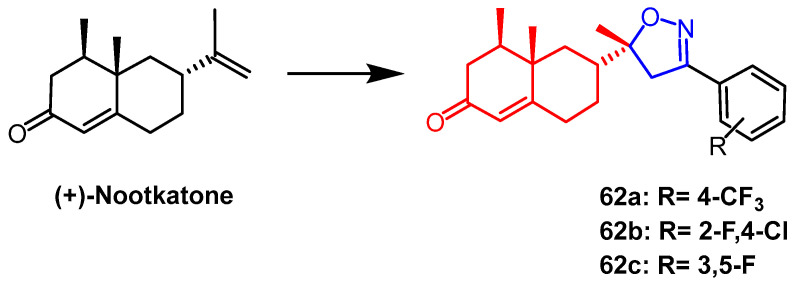
The chemical structure and derivatives of (+)-nootkatone [87].

**Figure 49 pharmaceuticals-16-00228-f049:**
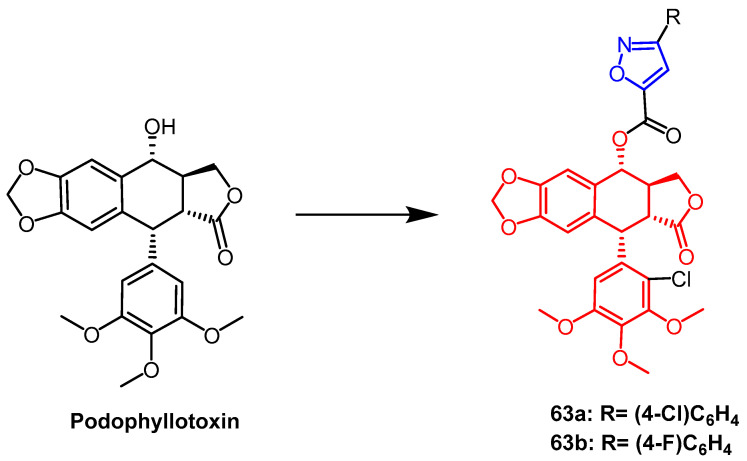
The chemical structure and derivatives of podophyllotoxin [88].

**Figure 50 pharmaceuticals-16-00228-f050:**
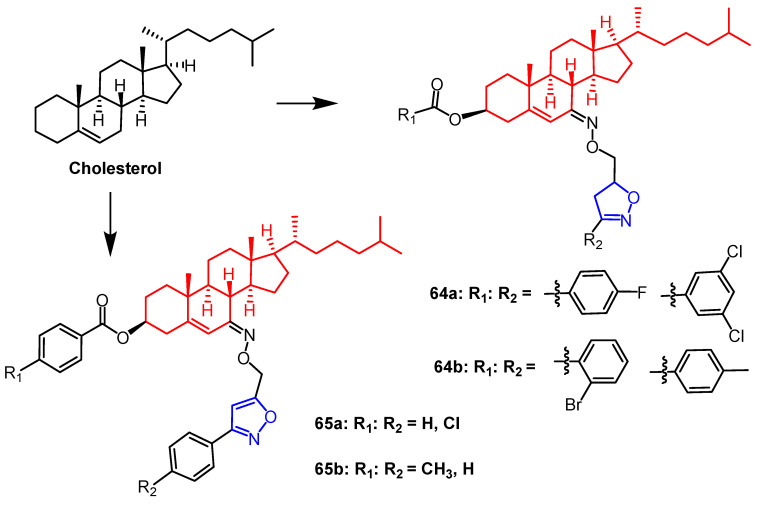
The chemical structure and derivatives of cholesterol [89].

**Figure 51 pharmaceuticals-16-00228-f051:**
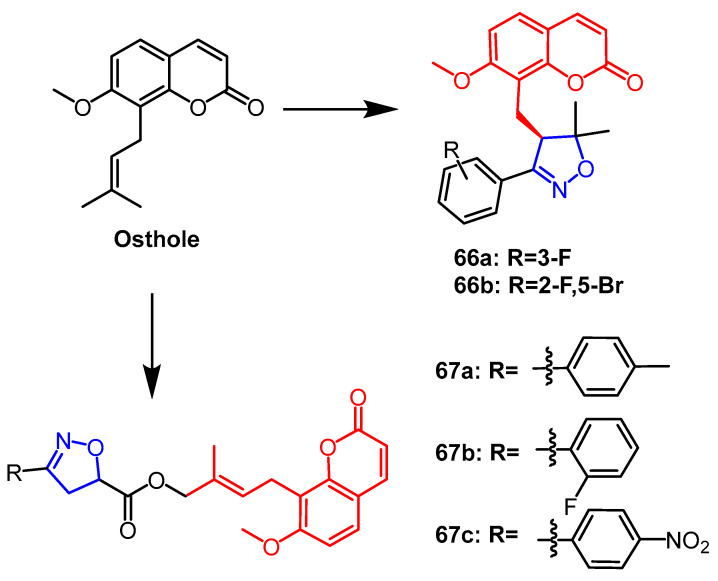
The chemical structure and derivatives of osthole [90,91].

**Figure 52 pharmaceuticals-16-00228-f052:**
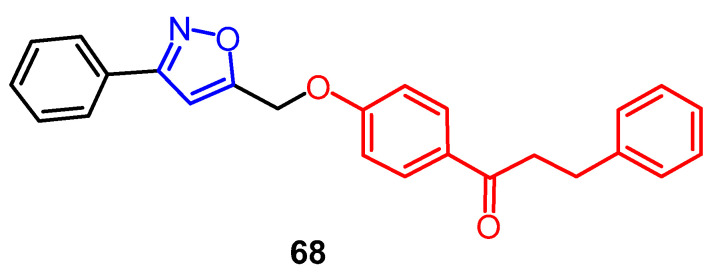
The derivatives of chalcone [92].

**Figure 53 pharmaceuticals-16-00228-f053:**
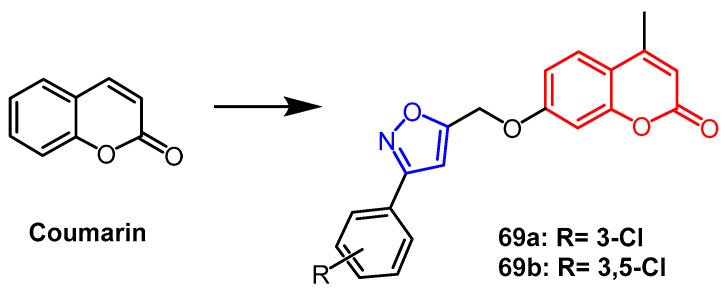
The chemical structure and derivatives of coumarin [93].

**Figure 54 pharmaceuticals-16-00228-f054:**
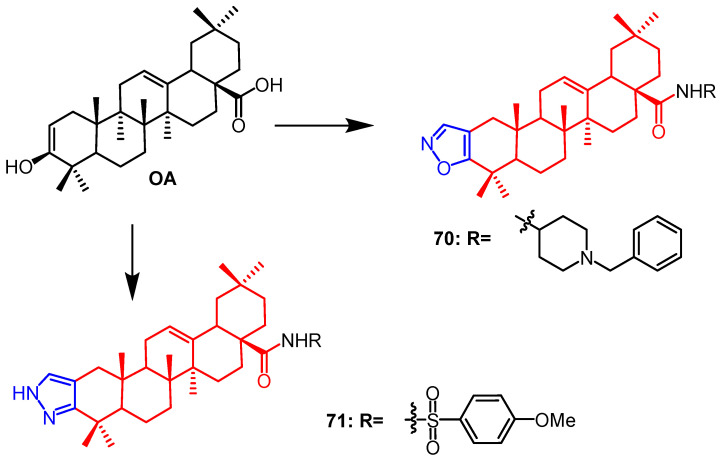
The chemical structure and derivatives of OA [94].

**Figure 55 pharmaceuticals-16-00228-f055:**
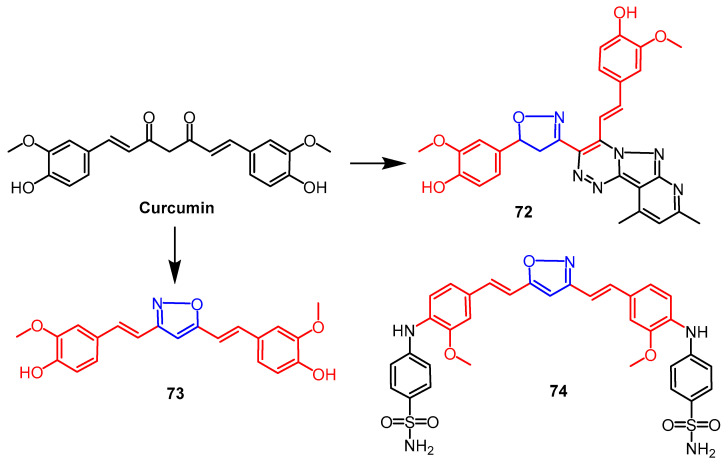
The chemical structure and derivatives of curcumin [95,96,97].

**Figure 56 pharmaceuticals-16-00228-f056:**
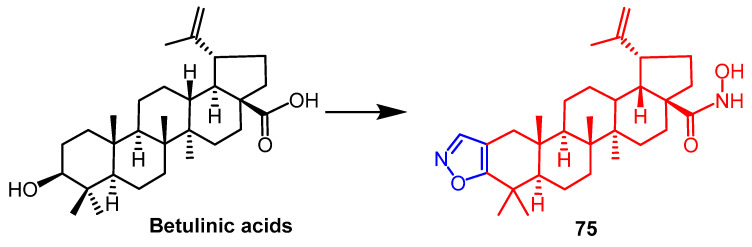
The chemical structure and derivatives of betulinic acids [98].

**Figure 57 pharmaceuticals-16-00228-f057:**
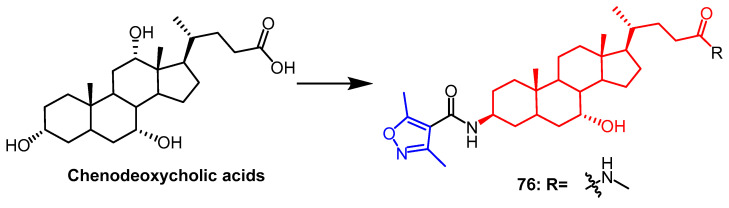
The chemical structure and derivatives of methyl chenodeoxycholic acids [99].

**Figure 58 pharmaceuticals-16-00228-f058:**
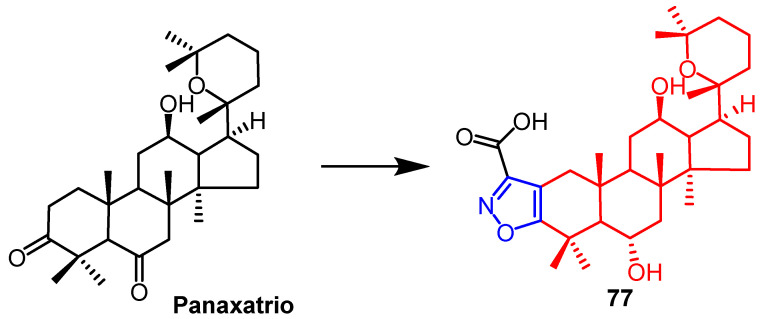
The chemical structure and derivatives of panaxatriol [100].

## Data Availability

Not applicable.

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
