# Peer review of "Isoxazole/Isoxazoline Skeleton in the Structural Modification of Natural Products: A Review"

_pharmaceuticals, 2023, doi:10.3390/ph16020228_

Round 1

Reviewer 1 Report

In this review article, the authors summarized the recent research progress of the isoxazole/isoxazoline skeleton in the structural modification of natural products. The applications of different natural product structures containing isoxazole/isoxazoline moiety in various fields are described in detail. The application of these structures in related fields can provide new conceptual methods and directions for future research. After reading the manuscript, I think it can be accepted for publication after major revision.

1. There are many blanks in the text (such as lines 463, 476, and 820), please check them carefully.

2. For each Figure, it is better to give the reference citation in the Figure caption.

3. The text after the comma in line 375 should be lowercase. Please check the full text.

4. The English should be polished.

5. The meaning of the structure marked in red and blue should be indicated in the text.

6. The conclusions and perspectives section should be streamlined.

7. Many abbreviations in the text are unnecessary.

Author Response

Response to Reviewer 1 Comments

Point 1: There are many blanks in the text (such as lines 463, 476, and 820), please check them carefully.

Response 1:We are very sorry for our error in writing the format. Thank you very much for pointing this out. We have removed the unnecessary spaces and checked the entire text.

Point 2:  For each Figure, it is better to give the reference citation in the Figure caption.

Response 2: Thank you very much for pointing this out. For each figure, we added a reference citation and copyright in the figure caption.

Point 3:  The text after the comma in line 375 should be lowercase. Please check the full text.

Response 3: We are very sorry that the grammatical details in the article have not been properly handled. Thank you very much for pointing this out. We have checked the full text and made changes.

Point 4: The English should be polished.

Response 4: Thank you for your valuable and thoughtful comments. We have carefully checked and improved the English writing in the revised manuscript.

Point 5: The meaning of the structure marked in red and blue should be indicated in the text.

Response 5: Thank you for your decision and constructive comments on my manuscript. We have explained the meaning of the structure represented by red and blue marks under each figure caption.

Point 6:  The conclusions and perspectives section should be streamlined.

Response 6: Thank you for your valuable and thoughtful comments.We have streamlined the conclusion and opinion sections by removing some unnecessary statements.

Point 7:  Many abbreviations in the text are unnecessary.

Response 7: Thank you for your valuable comments.We have removed some unnecessary abbreviations such as growth inhibition (GI), nitric oxide (NO), etc.

We tried our best to improve the manuscript and made some changes in the manuscript. These changes will not influence the content and framework of the paper. And here we did not list the changes but marked in red and blue in revised paper.

We appreciate for Editors and Reviewers’warm work earnestly, and hope that the correction will meet with approval.

Once again, thank you very much for your comments and suggestions.

Reviewer 2 Report

Authors summarize the importance of isoxazole and isoxazoline derivatives in several drug candidates which is highly convenient for researchers working in this topic. Some changes should be addressed before acceptance:

11)  Minor mistakes:

a) Authors should refer to other research groups by using their surname instead of names. For instance, lines 147, 206, 276, among others.

b) Line 154: remove – in 1,2,3-triazoles. Line 332: mistake in ortho positions.

c) Figures 29, 37 and 50 are identical. Please check this and draw isoxazole ring properly.

d) Figures description: “The chemical structure and derivatives of “x” results repetitive and should be changed in each case.

22)  Some Schemes also results repetitive: They are practically based on the already known molecule, arrow and subsequent modification with the isoxazole ring. Some variations in Schemes should be carried out.

33)  Most figures are really schemes. Please replace.

44)  In conclusions, lines 993 and 994 should be deleted. This is not really a synthetic review because there is not any mention about chemical transformations.

Author Response

Response to Reviewer 2 Comments

Point 1: Authors should refer to other research groups by using their surname instead of names. For instance, lines 147, 206, 276, among others.

Response 1:We are sorry for not double-checking the details of the manuscript. Thank you very much for pointing this out. We have changed all author names in the manuscript to surname. For instance, lines 709 Algethami et al., 656 sahoo et al., 485 Reddy et al.etc.

Point 2: Line 154: remove – in 1,2,3-triazoles. Line 332: mistake in ortho positions.

Response 2: Thank you for your valuable and thoughtful comments. We have removed the 1,2,3-triazoles and revised the ortho positions as suggested by the reviewers.

Point 3: Figures 29, 37 and 50 are identical. Please check this and draw isoxazole ring properly.

Response 3: We are sorry for the duplication of figures in the manuscript. Thank you very much for pointing this out. We have removed the duplicate figures and renumbered the figures in the full manuscript.

Point 4: Figures description: “The chemical structure and derivatives of “x” results repetitive and should be changed in each case.

Response 4: Thank you for your valuable and thoughtful comments. We have modified the duplicate images and results in the manuscript, for example, in Figure 23, and we have combined the two images into one and illustrated the results.

  • Point 5: Some Schemes also results repetitive: They are practically based on the already known molecule, arrow and subsequent modification with the isoxazole ring. Some variations in Schemes should be carried out.

Response 5: Thank you for your valuable and thoughtful comments. We have modified and removed the duplicate results and figures and presented the structural modifications of isoxazole and isoxazoline separately.

Point 6: Most figures are really schemes. Please replace.

Response 6: Thank you for your valuable and thoughtful comments. For each figure, we added a reference citation and copyright in the figure caption.

Point 7: In conclusions, lines 993 and 994 should be deleted. This is not really a synthetic review because there is not any mention about chemical transformations.

Response 7: Thank you for your valuable comments.We have removed two lines of text and streamlined the conclusions as suggested by the reviewers.

We tried our best to improve the manuscript and made some changes in the manuscript. These changes will not influence the content and framework of the paper. And here we did not list the changes but marked in red and blue in revised paper.

We appreciate for Editors and Reviewers’ warm work earnestly, and hope that the correction will meet with approval.

Once again, thank you very much for your comments and suggestions.

Reviewer 3 Report

Pan and co-workers diligently covered the review on Isoxazole/ isoxazoline skeleton in the structural modification of natural products. Isoxazole and isoxazoline are very important moieties present in many bioactive and natural products and drug molecules. More than 80 molecules with a broad range of bioactive are covered in the reported review. I believe it's a distinctive way to cover the review on natural products and drug molecules which gives the reasoning and background knowledge to readers. They also covered how the synthetic modification of this pharmacophore likely enhances the biological properties of the natural Pharmaceuticals of products, resulting in potent new molecules. I recommend manuscripts to publish in the given format. 

Author Response

Response to Reviewer 3 Comments

We appreciate for Reviewers’ warm work earnestly. We have tried our best to improve the manuscript and made some changes in the manuscript. These changes will not influence the content and framework of the paper. And here we did not list the changes but marked in red and blue in revised paper, and hope that the correction will meet with approval.

Once again, thank you very much for your recognition.
